# Combining resistivity and frequency domain electromagnetic methods to investigate submarine groundwater discharge (SGD) in the littoral zone

Marieke Paepen[1], Daan Hanssens[2], Philippe De Smedt[2], Kristine Walraevens[1], and Thomas Hermans[1]

[1]Laboratory of Applied Geology and Hydrogeology, Department of Geology, Ghent University, Krijgslaan 281-S8, Ghent, 9000, Belgium
[2]Research Group Soil Spatial Inventory Techniques, Department of Environment, Ghent University, Coupure links 653, Ghent, 9000, Belgium

*Correspondence to*: M. Paepen (Marieke.Paepen@UGent.be)

**Abstract.** Submarine groundwater discharge (SGD) is an important gateway for nutrients and pollutants from land to sea. While understanding SGD is crucial for managing nearshore ecosystems and coastal freshwater reserves, studying this discharge is complicated by its occurrence at the limit between land and sea, a dynamic environment. This practical difficulty is exacerbated by the significant spatial and temporal variability. Therefore, to capture the magnitude of SGD, a variety of techniques and measurements, applied over multiple periods, is needed. Here, we combine several geophysical methods to detect zones of fresh submarine groundwater discharge (FSGD) in the intertidal zone, upper beach, dune, and shallow coastal area. Both terrestrial electrical resistivity tomography (ERT, roll-along) and marine continuous resistivity profiling (CRP) are used from the shallow continental shelf up to the dunes, and combined with frequency domain electromagnetics (FDEM) mapping in the intertidal zone. Particularly we apply robust apparent electrical conductivity (rECa) estimation from FDEM data to provide reliable lateral and vertical discrimination of zones of FSGD. The study area is a strong dynamic environment along the North Sea, characterised by semi-diurnal tides between 3 and 5 m. CRP is usually applied in calmer conditions, but we prove that such surveys are possible and provide additional information to primarily land-bound ERT surveying. The 2D inversion models created from ERT and CRP data clearly indicate the presence of FSGD on the lower beach or below the low water line. This discharge originates from a potable freshwater lens below the dunes which flows underneath a thick saltwater lens, present from the dunes to the lower sandy beach, which is fully observed with ERT. Freshwater outflow intensity has increased since 1980, due to a decrease of groundwater pumping in the dunes. FDEM mapping at two different times reveals discharge at the same locations, clearly displays the lateral variation of the zone of discharge, and suggests that FSGD is stronger at the end of Winter compared to the beginning of Autumn. ERT, CRP, and FDEM are complementary tools in the investigation of SGD. They provide a high-resolution 3D image of the salt and freshwater distribution in the phreatic coastal aquifer over a relatively large area, both off- and onshore.

# 1 Introduction

The interface between land and sea is a complex environment, with groundwater discharging from the land to subterranean estuary and salt water intruding into coastal aquifers (Duque et al., 2020). Submarine groundwater discharge (SGD) and seawater intrusion (SI) are complementary processes, both subjected to the balance between hydraulic and density gradients in ground- and seawater (Taniguchi et al., 2002). SGD is a combination of fresh SGD (FSGD) and recirculated saline groundwater discharge (RSGD) (Taniguchi et al., 2002). It is driven by the terrestrial hydraulic gradient, water level differences at permeable barriers, wave set-up, tides, storm or current-induced pressure gradients, convection, seasonal fresh-seawater interface movement, bioturbation, and geothermal heating (Taniguchi et al., 2002 ; Michael et al., 2005 ; Burnett et al., 2006 ; Taniguchi et al., 2019). Its total flux is significant, since it occurs over large areas (Burnett et al., 2003), probably being more important to the oceanic budgets of nutrients, carbon, and metals than rivers (Moore, 2010).

FSGD can lead to eutrophication (e.g. Colman et al., 2004), blooms of harmful algae (e.g. Lapointe et al., 2005), and it can result in a significant loss of freshwater. RSGD can also have a significant impact on the dissolved nutrient input, when it interacts with the sediment (Rodellas et al., 2018). On the other hand, SI threatens potable water in coastal zones. Therefore, understanding the dynamics of salt and freshwater is essential for management of coastal freshwater reserves. However, assessment of SGD is difficult, because of spatial and temporal variability in flux (e.g. Burnett & Dulaiova, 2003 ; Michael et al.; 2005).

Geophysical techniques can be used to delineate salt and freshwater in coastal environments, facilitating the detection of FSGD and SI. Electrical and electromagnetic methods are particularly suited for spatial and temporal salinity delineation in coastal environments given their sensitivity to (water) electrical conductivity. Below we give an overview of the methods used in coastal research.

Time domain electromagnetic (TDEM) is mostly used for 1D soundings in coastal studies, for instance to measure salinity onshore (e.g. Goldman et al., 1991) or offshore, using a floating system (e.g. Goldman et al., 2004). Ground-based TDEM is time consuming and thus difficult to apply for 3D mapping. Similar, frequency domain electromagnetics (FDEM), although mostly used onshore, can be used offshore as well, by towing transmitter and receiver with a boat on the seafloor (Evans et al., 1999 ; Hoefel and Evans, 2001) or by keeping them above the water (Kinnear et al., 2013). FDEM is particularly suited for fast mapping of an area, but its depth resolution is limited and depends on the subsurface conductivity and the distance between the transmitting and receiving coils. The technique has been previously used to characterize pore water salinity in a coastal wetland (Greenwood et al., 2006), to map saltwater intrusion on Long Reef beach (Davies et al., 2015), combined with (continuous) vertical electrical and TDEM soundings to investigate the distribution of salt and freshwater on a sandy beach between Egmond aan Zee and Castricum aan Zee (Pauw, 2009), combined with 2D resistivity imaging to detect plumes of freshwater on a North Wales beach (Obikoya & Bennell, 2012), and combined with TDEM and onshore electrical resistivity tomography (ERT) to discriminate fresh and salt water in the Albufeira-Ribeira de Quarteira coastal aquifer system (Francés et al., 2015). However, when applying FDEM in saline environments, it is important to know that ground conductivity

instruments designed to operate under low induction number conditions (LIN) (e.g. CMD-MiniExplorer, DUALEM-421S, Geonics EM34, and EM31) provide apparent electrical conductivities (ECa) which are not valid in highly conductive environments (McNeill, 1980 ; Hanssens et al., 2019). This limitation is probably why FDEM has only rarely been applied in the intertidal zone and only the ECa values were used in the interpretation of these studies.

Both FDEM and TDEM methods are also available for airborne surveys, allowing to expand the coverage in coastal and tidal environments, see Siemon et al. (2009) and Viezzoli et al. (2012) for some examples. These surveys can be combined with for instance EM (e.g. Paine, 2003) or ERT studies on land (e.g. Viezzoli et al., 2010 ; Goebel et al., 2019).However, airborne surveys are costly and often aiming at large-scale structures, so they might lack resolution at shallow depth, which is crucial to image local FSGD.

Next to EM, ERT is another useful geophysical method in mapping coastal salinity. ERT allows to image the subsurface resistivity in 2D cross sections of 3D, after the data is inverted. The depth of investigation and resolution of ERT are respectively proportional and inversely proportional to the electrode spacing, but also depend on the distribution of the subsurface resistivity and the chosen electrode array (Dahlin & Zhou, 2004). The ERT cables and electrodes remain fixed during the measurement, so that it can be used above the high tidemark (e.g. Goebel et al., 2017) and in the intertidal zone to

measure for instance the tidal effect on groundwater exchange (e.g. Dimova et al. 2012). Time-lapse ERT is popular for monitoring SI (e.g. de Franco et al., 2009 ; Ogilvy et al., 2009), including cross-hole ERT to increase resolution (e.g. Palacios et al., 2019).

Resistivity surveying can be extended to water bodies by resting or burying the cables in the sediments (marine electrical resistivity – MER) (e.g. Swarzenski et al., 2006 ; Taniguchi et al., 2007 ; Swarzenski & Izbicki; 2009 ; Henderson et al., 2010).

In case of saline water, MER provides a better resolution, since the conductive water layer on top has less effect on the signal compared to continuous resistivity profiling (CRP), where the current has to pass the water layer before entering the sediment. In addition, MER allows to collect reciprocal data to estimate the noise level. It is more time consuming, but best suited for monitoring. In contrast, by using CRP, one can easily cover several kilometres per day. In order to compensate the effect of the water layer (between the streamer and sediment) during the inversion process, the bathymetry and water conductivity are

measured at the same time. The resolution of the measurements depends on the electrode spacing, height of the waves, and the boat speed. Most CRP studies found in literature were carried out in quiet environments with limited tidal and wave activity such as Biscayne Bay (Swarzenski et al., 2004), Neuse River Bay (Cross et al., 2006), Florida Bay (Swarzenski et al., 2009), Manhasset Bay and Northport Harbor (Cross et al., 2012), and Great South Bay (Cross et al., 2013), Jobos Bay in Puerto Rico (Day-Lewis et al., 2006), and the fringing reef of Santiago Island in the Philippines (Cardenas et al., 2010 ; Cantarero et al.,

2019). These are sometimes combined with Radon-222 (e.g. Swarzenski et al. 2004 ; 2009), MER (e.g. Swarzenski and Izbicki, 2009), seismics (e.g. Russoniello et al., 2013 ; Cross et al., 2014), EM logs in temporary boreholes (e.g. Krantz et al., 2004) or salinity measurements (Cardenas et al., 2010 ; Cantarero et al., 2019).

Most of the geophysical studies identified above do not bridge the gap between the land and marine realm. Offshore and onshore surveys are rather performed separately, especially if conditions are difficult, e.g. strong tidal and wave activity.

Additionally, the spatial and temporal variability of FSGD make it more challenging. Onshore surveys alone are generally not sufficient to image the full extent of the FSGD zone, while offshore surveys miss the necessary connections with onshore freshwater aquifers. Similarly, while FDEM surveys allow for faster mapping at higher resolution of the salinity than resistivity methods, they do not bring sufficient information on the vertical distribution of salinity.

In this paper, we combine for the first time ERT, CRP, and FDEM. The objective of the survey is to qualitatively map the lateral and vertical distribution of salt and freshwater and identify the FSGD zone underneath the near-shore continental shelf, intertidal zone, upper beach, and dunes, at 'De Westhoek' nature reserve, Belgium. ERT and CRP provide changes of the subsurface resistivity in 2D cross-sections, while FDEM mapping with robust apparent electrical conductivity (rECa) provides additional information on lateral variations in the intertidal zone between parallel cross-sections. The combination of CRP

during high tide and land ERT collected at low tide ensures that FSGD and SI can be accurately imaged at the limit of the low water line. Our case study demonstrates that only a combination of the proposed methods allows to fully characterize the discharge zone in the target area. It also shows that CRP data can directly provide reliable identification of the presence of fresher water, even in very dynamic environments with a large tidal range, and that FDEM monitoring can be used to quickly assess seasonal variations in FSGD in the intertidal zone.

## 2 Study area

The study site is located in the most western part of the 65 km long sandy coastline of Belgium, which is delimited by the North Sea in the North and a SW-NE oriented, semi-continuous dune belt in the South, which varies in width. These dunes are part of a long, narrow dune strip that runs from Calais (France) to Northern Denmark. The Belgian coastal dune belt is around 2 km wide at 'De Westhoek' – an extensive Flemish nature reserve and European Natura 2000 area – which is located between

20 the Belgian-French border and the city of De Panne (Figure 1). A groundwater extraction site is situated in the dunes, exploited by Intercommunale Waterleidingsmaatschappij van Veurne-Ambacht (IWVA). Potable groundwater has been pumped there since 1967.

A low-lying polder area borders the dunes in the South. The shore in the North is tidal dominated (semi-diurnal), with a tidal range of over 5 m at spring tide and approximately 3 m at neap tide. The beach is relatively flat with an average slope of 1.1%

(Lebbe, 1981). The shore morphology, tidal range, and permeability of the phreatic aquifer determine the presence of a saltwater lens under the shore – salt water can infiltrate on the back shore during high and spring tide and discharges on the fore shore and/or seabed (Vandenbohede & Lebbe, 2006).

The sandy aquifer is around 30 m thick at the shore and up to 50 m in the dunes (due to a recent aeolian sand cover) and bounded by Late Eocene heavy marine clays of the Kortrijk Formation, which are considered impermeable. The aquifer is

30 relatively homogeneous at the Belgian-French border, the upper part is fine medium sand and the lower part more coarse and medium sands. It becomes more heterogeneous towards the East, with the local occurrence of clayey/silty fine sand lenses (e.g. Lebbe, 1978, 1999). For instance, a clay lens occurs between -5 and -10 mTAW (compared to the mean sea level at low

tide in Ostend, which is the Belgian reference level) under the upper beach and fore dunes in the East of the study zone (Vandenbohede et al., 2008b; Hermans et al., 2012).

Freshwater recharges in the dunes, forming a large freshwater lens underneath. Part of this freshwater flows underneath the large saltwater lens, forming a freshwater tongue, discharging into the North Sea around the low water line (Vandenbohede & Lebbe, 2006). The potable water is of high value in the coastal region, since salt and brackish water are often found at shallow depth (i.e. the beach and polder area) (De Breuck et al., 1974 ; Delsman et al., 2019). The scarcity of freshwater in the region and the high demand of drinking supplies, especially in Summer (due to coastal tourism), make a comprehensive understanding of the hydrogeological system important for the sustainable management of the phreatic groundwater reserves.

In the past, the migration of fresh groundwater towards the sea has been observed, at 'De Westhoek', by electrical well logging in 1980 (Lebbe, 1981, 1999) and temperature measurements (Vandenbohede & Lebbe, 2011). Several hydrogeological and groundwater flow models have been developed showing sensitive parameters and past, present, and future evolution of the freshwater tongue (Lebbe, 1983, 1984, 1999 ; Lebbe & Walraevens, 1988 ; Van Camp & Walraevens, 2004 ; Vandenbohede & Lebbe, 2006, 2007 ; Vandenbohede et al., 2008a). However, these models are 2D cross-sections and do not account for the lateral variations along the coast line. The latter can be revealed by geophysics. In addition, conditions have changed over the years, since the IWVA started a decrease of the exploitation rate at 'De Westhoek' in the 1990s. Nowadays, the pumping rate (approximately $3.2 \times 10^5$ m³ in 2018) is much smaller compared to the 1980s, when pumping rates were between 1.5 and $2.2 \times 10^6$ m³ yr$^{-1}$.

## 3 Methodology

### 3.1 Electrical resistivity

#### 3.1.1 ERT on land

##### 3.1.1.1 Data acquisition

The ABEM Terrameter LS with 64 electrodes was used for the ERT survey on land. Data acquisition took place on multiple days – 7 March (one profile 1 km East of the Belgian-French border, K1) and 11 October 2018 (one profile at the border, K0). Measurements started during the lowest tide, allowing to measure a wide zone, with the roll-along technique. Using this method, the Terrameter is placed between electrodes 32 and 33. After measuring, it is shifted by 16 or 32 electrodes while new electrodes are positioned at the end of the profile so that the device is back in the middle of the array to perform new measurements. This procedure can be repeated numerous times, to obtain very long ERT profiles, while maintaining relatively homogeneous lateral sensitivity. It also allows to maintain all the measuring electrodes above the current tide level. Combined with a spacing of 5 m between the electrodes, profiles of up to 625 m long were obtained. The multiple-gradient array configuration was chosen for its relatively fast data acquisition speed, which is critical in tidal-dominated environments, and its good compromise between signal-to-noise ratio and resolution (e.g. Dahlin & Zhou, 2004, 2006).

### 3.1.2 Marine CRP

#### 3.1.2.1 Data acquisition

For the marine survey, direct-current CRP is preferred over MER mode (in which electrodes are buried or rest on the sea bottom), since data collection is easier and faster. The use of CRP in a very dynamic environment, such as the North Sea (semi-diurnal tides - tidal range between approximately 3 and 5.3 m - and often relatively strong waves), is not common. Measurements were only performed under relatively calm weather conditions, maximum wave height around 1 m and wind speed maximal 4 bft. CRP is performed near-shore, both during high and low tide. The former ensures some overlapping with ERT on land and the latter improves the resolution offshore, since the seawater layer is thinner in the low-tide conditions.

Water-borne data was collected at high tide during a single day in 2018 (30 May). A total of two perpendicular and three parallel profiles were obtained. The Syscal Pro Deep Marine (IRIS Instruments) was used, combined with a 195 m long floating streamer of 13 graphite takeouts – 2 fixed current electrodes and 11 potential electrodes – spaced at a 15 m interval. Using a reciprocal Wenner-Schlumberger configuration, with the current electrodes in the middle of the set-up, the simultaneous collection of 10 potentials was obtained for each current transmission (of 35 to 37 A). The streamer was towed at a relatively constant speed (around 3.5 km h$^{-1}$) by a small rigid inflatable boat of the Flanders Marine Institute (VLIZ) collecting data continuously. In order to correct for the influence of the highly conductive seawater layer on the signal, a Garmin GPSMAP 188 Sounder system was directly connected to the Syscal unit (obtaining coordinates and water depth) and seawater conductivity was separately recorded using a CTD diver.

An additional marine survey was carried out on 22 May 2019, allowing to measure three perpendicular profiles during low tide in front of 'De Westhoek'. Again, the Syscal was used with the same set-up, but with a cable of 130 m. The 13 electrodes were spaced at 10 m, allowing a better resolution yet sufficient penetration in the seabed.

#### 3.1.2.2 Pre-processing

In a first step, the raw marine data was checked showing the general trend in apparent resistivity. For the perpendicular high tide profiles, e.g. K0$_{HT}$ (Figure 2), an increase in resistivity is observed towards the beach. This is not only caused by the decreasing thickness of the seawater layer – since other perpendicular CRP profiles made at Wenduine (Belgium) (where possibly no or much less FSGD occurs, Figure 2) barely show this trend towards the land – but mainly due to the outflow of fresh or brackish groundwater. The raw data also displays two bad acquisition channels (rho 6 and rho 7, Figure 2) in all marine profiles (2018) caused by one damaged electrode. These channels were removed from the datasets, reducing the sensitivity of the inversion models.

The bathymetry was filtered to remove the noise effect caused by the waves, by averaging the data. The echosounder measurements of 2018 also comprise multiples. These were calibrated with the known shore topography.

### 3.1.3 Processing

Both marine and land ERT data were inverted using the RES2DINVx32 ver. 3.71.118 software. For marine data, potentials were continuously measured which prevents the estimation of the error through stacking or reciprocal measurements. For land ERT, reciprocals could not be collected because of time constraints related to the rising tide. The repeatability error was low, indicating data of good quality. For the CRP inversion, the water layer was included using the measured seawater conductivity (approximately 5 080 mS m$^{-1}$) and bathymetry (Loke, 2011). Note that the conductivity of the water layer is introduced as a soft constraint and is not fixed during the inversion. This option was used to prevent the creation of artefacts of inversion as can occur when (potentially erroneous) strong constraints are introduced (Henderson et al., 2010 ; Caterina et al., 2014). Bad data points were filtered after a first inversion based on the root mean square error ($E_{RMS} > 100\ \%$). A robust inversion was carried out, using the floating electrodes survey option for the marine profiles. Given the absence of robust error estimates, we relied on the convergence criterion typically used for ERT to stop the inversion process (decrease of $E_{RMS}$ lower than 0.1 % between two iterations). The mean absolute error of all inverted models lies below 2.8 %.

### 3.1.4 Inversion model appraisal

The resolution of resistivity profiles quickly decreases with depth. This is particularly true for marine profiles as a large portion of the current directly flows in the water layer. The inversion of marine data is thus subject to large uncertainty, especially, the quantitative interpretation of inversion results might be difficult (Day-Lewis et al., 2006). In our case, we are mostly interested in the relative variation of resistivity indicating the presence of fresher water. To obtain information on how much the model is influenced by the inversion parameters (bathymetry, starting model, etc.) and if reliable resistivity variations are mapped, we propose an approach based on the two-sided difference developed by Oldenburg and Li (1999) for estimating the depth of investigation (DOI) index. The DOI is an image appraisal tool which is often used to indicate which portions of the inversion model can be interpreted. It is calculated based on two additional inversions ($\rho_{inv,1}$ and $\rho_{inv,2}$) using the same dataset, but for which 0.1 and 10 times the reference resistivity ($\rho_{ref,1}$ and $\rho_{ref,2}$) are imposed as a reference model:

$$DOI = \left| \frac{\log \rho_{inv,1} - \log \rho_{inv,2}}{\log \rho_{ref,1} - \log \rho_{ref,2}} \right| \quad (1)$$

A low DOI value is obtained when resistivity structures in the model are driven by the data and not by the inversion process which is influenced by the reference model. Although a threshold between 0.1 and 0.2 on the DOI value is often used to delimit reliable zones of the image, this choice is subjective (Caterina et al., 2013). In addition, the DOI is directly sensitive to the absolute value of resistivity and can thus yield high values even for consistently resistive or conductive structures. Therefore, we do not interpret the absolute DOI value itself, even though it is very low throughout most inversion models, but interpret it qualitatively with the three respective inversions. We assume that we can make a robust qualitative interpretation of the inversion model, when all three inverted models show similar resistivity patterns. We are more careful in the evaluation if the

models do not agree on the resistivity of certain zones. For marine data, the DOI is especially influenced by the thickness of the seawater layer (bathymetry), which has a large effect on the calculated value.

## 3.2 Electromagnetic

### 3.2.1 Data acquisition

Two different FDEM devices were used at 'De Westhoek'. Both have a short acquisition time and are relatively easy to use in the intertidal zone. The DUALEM-421S has the advantage of a larger investigation depth compared to the CMD-MiniExplorer, which only gives information of the upper 2 m.

The CMD-MiniExplorer (GF Instruments, s.r.o., Czech Republic) is a portable multi-depth FDEM device which measures apparent electrical conductivities (mS m$^{-1}$). It was used for mapping on 23 February 2018. The unit is relatively small and,

therefore, easy to use. The entire device is operated by one person, who carries the 1.62 m long probe together with a Trimble GPS (Trimble Navigation Ltd., Sunnyvale, California, USA). The system operates at 30 kHz and contains three different receiver coils (dipole distances of 0.32 m (CMD1), 0.71 m (CMD2), and 1.18 m (CMD3)), horizontal coplanar (HCP) configuration was used, giving a cumulative sensitivity of respectively 0.5, 1.0, and 1.8 m under LIN (Callegary et al., 2007). Multiple-receiver FDEM data were recorded using a DUALEM-421S instrument (DUALEM Inc., Milton, Canada) mounted

on a sled and towed by a quad on 27 and 28 September 2018, after a very warm and dry summer. The device worked in HCP mode, which resulted in three HCP configurations with a coil spacing of 1 (HCP1), 2 (HCP2), and 4 m (HCP4), and three perpendicular (PRP) configurations with a coil spacing of 1.1 (PRP1), 2.1 (PRP2), and 4.1 m (PRP4). Therefore, six different volume-related apparent conductivities are recorded simultaneously. The depths at which the signal sensitivities are highest is ambiguous, since these depend on the specific subsurface conductivity distribution. The approximate depths of investigation

provided by the six coil configurations are 1.5, 3, 6, 0.5, 1, and 2 m respectively, but have no physical meaning in this highly conductive environment. To allow straightforward comparison between datasets obtained with different coil configurations and instruments, these are here referred to as pseudodepths. The instrument's operating frequency was 9 kHz and elevation above the ground was 0.195 m. The sampling frequency was 8 Hz at an acquisition speed of approximately 8 km h$^{-1}$, rendering an in-line sampling spacing of approximately 0.25 m. Survey lines were repeated in parallel lines at a spacing of approximately

5.5 m. Geographic coordinates were logged using a Trimble R10 GNSS system (Trimble Navigation Ltd., Sunnyvale, California, USA).

### 3.2.2 Processing

Basic processing of the DUALEM measurements was done following Delefortrie et al. (2014b, 2015). The linear relation between the quadrature-phase (QP) component of the electromagnetic field and the LIN ECa (McNeill, 1980) – used in most

commercially available FDEM equipment – is not valid in highly conductive environments, as it generally underestimates electrical conductivity (Figure 3, left). Different from most studies, we converted the LIN ECa data to robust ECa (rECa),

enabling qualitative FDEM data interpretation in a highly conductive environment. We follow the non-linear approach of Hanssens et al. (2019), in which rECa (mS m$^{-1}$) data are calculated from the QP data to accurately estimate ECa at higher induction numbers (Figure 3, right). Ultimately, the data were interpolated to a 2 by 2 m grid, via natural neighbour interpolation.

## 3.3 Interpretation

The inversion of resistivity data acts as a filter which tends to blur the obtained image: the inverted resistivity is only an estimation of the true resistivity. For this reason, it is difficult to directly relate an inverted resistivity value to a specific value of the salinity or total dissolved solid content (TDS). However, in this case, we have two well-identified extremes (freshwater in the dunes and seawater), together with a petrophysical model developed by Lebbe (1978, 1981, 1999) for the Western Belgian coastal plain, allowing a semi-quantative interpretation of resistivity. For sandy sediments at 'De Westhoek', Lebbe (1981) estimated the formation factor of Archie's law (Archie, 1942), F = 3.2:

$$\rho_w = \frac{\rho_b}{F} \ (2)$$

where the bulk resistivity ($\rho_b$), is deduced from ERT/CRP profiling and FDEM mapping (given the scale of measurement, we assume that the rECa can approximate $\rho_b$) and $\rho_w$ is the pore water resistivity. Here, we propose a semi-quantitative interpretation in salinity classes based on relative variations in resistivity (and conductivity). Due to measurement errors, resolution, and inversion constraints, deducing the effective TDS would only be possible with a specific calibration of geophysical measurements based on ground truth data. We distinguish three main water quality classes (salt, brackish, and fresh) to interpret the geophysical data in the specific study area (De Moor and De Breuck, 1969).

## 4 Results and discussion

In the following paragraphs, we present the data and results for each of the used methods, starting with FDEM, then land and marine resistivity measurements. In the East of the study area, FDEM data were acquired during two different seasons, allowing to investigate seasonal variability of the fresh groundwater discharge. By comparing our results to previous observations, we can see how the fresh-/saltwater distribution has changed since the 1980s, due to a decrease of pumping in the nearby groundwater extraction facility. In the following, we will use a unique colour scale to describe the results in all figures (except in Figure 4): freshwater occurs in zones with a resistivity higher than approximately 20 Ωm (blue), salt water has a resistivity lower than roughly 2.5 Ωm (red), and brackish water leads to intermediate resistivities (light brown to light blue). In the case of the Belgian coast, the strong tides play a role allowing salt water to penetrate in the shallow sediment, and making the detection of FSGD - based on resistivity/conductivity - more difficult. The discharge zone will thus not be necessarily characterized by freshwater but rather by brackish water.

## 4.1 FDEM

We started our investigation with the CMD-MiniExplorer mapping at the location of one of the profiles of Lebbe (1981), K1, located at 1 km East from the French-Belgian border. The CMD-MiniExplorer mapping (Figure 3, right, and Figure 4, A.) identifies the presence of FSGD close to the low water line, at K1, indicated by a decrease in the electrical conductivity. The FDEM data clearly demonstrates a zone which is over 100 m wide. The outflowing water is very brackish to moderately salt, the conductivity is roughly between 350 and 650 mS m$^{-1}$, due to a mixture of discharging fresh groundwater and seawater that infiltrated on the beach during high tide.

To investigate lateral variation between the western and eastern part of the study area, the intertidal zone was also mapped with the DUALEM at the vicinity of the low water line. This map clearly demonstrates a northward shift of the zone of discharge from East towards West (Figure 4, B.-D.). From approximately 300 m East of the border, the discharge zone is no longer visible from the EM data since the discharge is located below the low water line. The DUALEM was dragged on the beach in a sled, making the quality of the data higher compared to the MiniExplorer, which was carried by hand, making it difficult to maintain a constant height above the surface during mapping. The water conductivity, obtained with the DUALEM-421S, in the discharge zone corresponds to a brackish water type, confirming the mixing of freshwater with salty water.

## 4.2 Land ERT

Interpreted on its own, the conductivity variations observed with FDEM could be related to lithological heterogeneities or the presence of near-surface features. Therefore, long land ERT profiles (Figure 5) covering the entire zone between the low water line and the dune area were collected. In the East (Figure 5, K1), we identify the freshwater aquifer located in the dune area. The brackish water observed at -10 mTAW corresponds to remnants of seawater infiltration during the flooding of an artificial tidal inlet, which started in 2004 and ended a few years ago. The downward movement of the denser seawater is hindered by the presence of a local clay lens, a process that was closely monitored in previous studies (Vandenbohede et al., 2008b; Hermans et al., 2012). Along this profile, there is a large saltwater lens on the beach. Underneath, freshwater is flowing from the dunes towards the North Sea. The freshwater mixes with the salt water, leading to the discharge of salt to brackish water during low tide on the lower beach. This is clearly visible on the land profile, close to the low water line, where fresher water is seen near the surface, which corresponds to the zone identified by FDEM measurements.

In the centre of profile K1 (Figure 5), the groundwater appears to be brackish. Here, the thickness of the salt water lens is maximum (15 m), while the bottom boundary of the aquifer (thick clay layer) is located between -25 and -30 mTAW. The lower resistivity is likely a result from the smoothness constraint inversion combined with the lower resolution at depth as the more resistive freshwater is bounded by two conductive layers, leading to a lower inverted resistivity (Hermans & Paepen, 2020).

There is a similar situation 1 km to the West, at the Belgian-French border (Figure 5, K0), where freshwater moves underneath a saltwater lens from the dunes towards the sea. Nevertheless, a striking difference on this profile is the absence of freshwater

flowing upwards and discharging on the lower beach. The saltwater lens, although becoming thinner towards the low water line, remains continuous. At the seaside of the profile, the underlying water is identified as fresh whereas it seems more brackish towards the dunes. This is probably related to the higher resolution at depth when the conductive salt layer is thinner and not due to an actual variation in salinity. On this profile, the clay layer (Kortrijk Formation) underlying the coastal aquifer

is detected around -35 mTAW. The ERT at the border gives the natural distribution of salt and freshwater, since the groundwater system is least affected by anthropogenic effects (extraction facility) in this part of the study area. As a consequence, the pore water is more resistive underneath the beach and further offshore compared to K1.

## 4.3 CRP

The land ERT profiles were extended by marine continuous resistivity profiles collected at high tide, with an overlapping zone

of about 230 m (Figure 6, $K1_{HT}$ and $K0_{HT}$). They confirm the presence of brackish water below the salt water. And indicate that the discharge zone is not limited to the low water line, but that it could extend towards the sea. In the East, weakly salt water is migrating towards the seabed around 200 m in front of the low water mark, (Figure 6, $K1_{HT}$, light red colour). Further offshore, a marine profile, collected at low tide (Figure 6, $K1_{LT}$), confirms that brackish pore water can be seen up to at least 550 m from the low water line. It is mixed with salt water in the seabed, making it impossible to visualize FSGD at the top of

the seabed using resistivity measurements, but the brackish groundwater is found relatively close to the seabed.

In the West, brackish groundwater is overlain by a thin layer in which pore water is salty (Figure 6, $K0_{HT}$). But the underlying brackish pore water is found closer to the seabed approximately 250 m offshore the low water line. The brackish lens (Figure 6, $K0_{LT}$) does not reach as far offshore compared to $K1_{LT}$, perhaps due to different local hydrogeological and geological conditions.

To acquire information on the lateral variation, CRP profiles parallel to the shore were collected during high tide. On the lower beach, the pore water resistivity increases from K1 towards K0 (Figure 7, M1). This trend is partly due to the thinner saltwater lens in the West, which increases the resolution of the inversion model. Modelling studies have shown that the saltwater lens thickness is inversely related to the groundwater flux (Vandenbohede and Lebbe, 2006). Part of the M1 profile (close to the middle) has a thicker lens compared to the zone around K1, while the water underneath K1 is more brackish, meaning that the

fresh groundwater flux is lower around K1. Also closer to the low water line, resistivity increases from K1 to K0 (Figure 7, M2). East of K1, a more resistive body (light blue) is seen (Figure 7, M3), also visible on the raw data. While in the far East of the M3 profile (in front of the IWVA site), the resistivity is much lower compared to K1, which is compatible with a lower fresh groundwater flux (Vandenbohede and Lebbe, 2006). The higher resistivity between the IWVA and K1 can have multiple reasons, which need further investigation. The thickness of the overlying saltwater lens can have an effect, as well as the width

of the beach (which is narrower East of K1). Local hydrogeological heterogeneities can have an influence too, since local clay lenses are present in the phreatic aquifer, which are difficult to identify in a saline environment due to their low resistivity. Maintaining the cable exactly parallel to the beach was also challenging, some measurement distortions are thus also possible.

To confirm the lateral variations in FSGD along this part of the coast, another perpendicular profile was collected further East (Figure 6, K1.5$_{LT}$). Discharge seems stronger on K1$_{LT}$ compared to K1.5$_{LT}$, since the brackish aquifer extends further offshore. This observation is logical, K1.5$_{LT}$ is closer to the IWVA extraction site and, so, more affected by the water extraction reducing groundwater discharge. However, this should be further examined.

## 4.4 Quality appraisal

The resolution of the high tide marine profiles are significantly lower compared to those acquired during low tide, since: it has a thicker seawater layer above the sea bottom, the portion of salt water infiltrating in the seabed increases from low to high tide, it is collected in other dynamic conditions (significant groundwater discharge might only occur at low tide, since the hydraulic gradient between land and sea is larger compared to the high tide), the electrodes of the streamer were spaced closer during the low tide survey (10 m compared to 15 m for the high tide profiles), and less channels were obtained in the high tide survey (due to a bad electrode).

To validate the results of the ERT and CRP inversions, we use the two-sided difference approach and the DOI, additionally computing the inversion for two different reference models. We show them for K1 (Figure 8), but other profiles have similar results. For the land profile, all three inversions (Figure 8, B1-3) look almost perfectly similar, with practically all DOI values below 0.1 (Figure 8, B4), indicating that observed structures are qualitatively contained in the data. For the profile at high tide (Figure 8, A1-3), lateral variations are also similar in the different inversion models, identifying fresher pore water towards the beach. However, small variations are observed vertically around the sea bottom, corresponding to DOI values above 0.2 (Figure 8, C4), showing that the inversion is particularly sensitive to the bathymetry and the presence of the seawater layer. In this case, a low conductivity reference model will force some more resistive features to appear closer to the sea bottom. As a consequence, brackish pore water seems closer to the sea bottom. Whether this is actually the case or not cannot be elucidated from our measurements and the true extension of the FSGD zone cannot be delimited with certainty from the high tide profile. For the low-tide profile (Figure 8, C1-3), fewer differences are observed compared to those taken during high tide and the discharge zone is more clearly identified. This is coherent with the higher resolution of the low-tide profile related to the thinner seawater layer. In summary, all profile types, although with different intrinsic resolutions, are sensitive enough to characterize the general salinity distribution in the studied zone.

## 4.5 Seasonal variations

The MiniExplorer and DUALEM data were collected in two different seasons, respectively the end of Winter and beginning of Autumn. The FSGD intensity around K1 is different in February (MiniExplorer) compared to October (DUALEM-421S), but the zone of discharge does not move (Figure 4). During the former period, the outflow of groundwater is clearly seen around K1 at shallow depth (Figure 4, A.). Whereas in October, a large zone of salty-brackish pore water is well visible at larger depth (Figure 4, D.), but only seen as individual spots of slightly lower conductivity in the shallow subsurface (Figure 4, B. and C.). According to the modelling results of Vandenbohede and Lebbe (2006), this seems to indicate that FSGD is stronger

in Winter than Autumn. This is likely due to the lower precipitation and higher evapo(transpi)ration in Summer, leading to smaller groundwater fluxes. In 2018, Summer was particularly dry in Belgium.

## 4.6 Long-term evolution

Previous studies had shown the presence of freshwater under the saltwater lens in this study area, but could not identify the discharge zone. The identified saltwater lens at K1 is similar in size as observed in 1980, based on well loggings on the beach (Lebbe, 1981) (Figure 9, top). However, the lens does no longer extend to the sea, since the freshwater tongue reaches the surface nowadays on the lower beach. This is probably an effect of the decreasing pumping trend of the IWVA exploitation site: the rate is now over 4.5 times lower compared to 1980, which has subsequently strengthened the groundwater discharge. Another explanation could be that the logs collected in 1980 were not sufficient to detect the discharge zone, that would have occurred between the two logs closest to the low water line, in which case the salt water lens would have remained relatively stable. However, this interpretation does not seem compatible with the newly acquired CRP data which identify the presence of freshwater further offshore. The shape and thickness of the saltwater lens at K0 have not changed much since 1980 (Figure 9). The lens becomes thinner towards the low water line, but there is no discharge on the lower beach, since FSGD is located below this level. K0 is located furthest from the IWVA site and no large extractions are known to occur on the French side of the border, so the effect caused by the decrease of the pumping is more limited.

## 4.7 Advantages of the proposed methodology

By combining ERT, CRP, and FDEM, we have mapped FSGD in 3D over a relatively large area. Which comprises the dunes, upper beach, intertidal zone, and part of the shelf. FDEM has allowed to image the lateral variation of the pore water conductivity in the upper meters of the intertidal zone, while ERT shows the vertical distribution of fresh and salt water, which is extended offshore thanks to CRP. Those results could not have been obtained with one of the technique alone, or even with the combination of two of them. The previous conceptual model of the study area (Hermans et al., 2012) can be updated and extended offshore (Figure 10).

For FDEM, the use of a robust estimation of the apparent conductivity (rECa) is crucial in saline environments. Classical estimations with the LIN approximation would underestimate the ECa and reduce the observed contrasts, likely leading to wrong interpretation in terms of salinity and FSGD zone. Using this appropriate correction, we can more easily compare FDEM with ERT, what is crucial in combined geophysical surveys.

It is interesting to note that the presence of fresher water is clearly visible on the raw CRP data as a gradual increase in the apparent resistivity (Figure 2). The apparent resistivity is the resistivity a homogeneous earth should have to provide with the same potential measurements and the same electrode configuration. It is not only influenced by the aquifer resistivity, but also (and mainly) by the low seawater resistivity, explaining why apparent resistivity remain low, even in presence of freshwater in the aquifer. The apparent resistivity corresponds to a specific combination of four electrodes. With a spacing of 10 or 15 m between the electrodes, this corresponds to an investigated volume with a minimum of 40 or 60 m in length. Therefore, the

transition caused by changes in the aquifer resistivity will be smoothed by the volume investigated. A sharp fresh/salt water interface will thus appear as gradual in the raw data. Yet, possible zones of FSGD can be identified without inversion. If the seawater influence remains constant (which is true as long as its thickness remains similar), an increase of the apparent resistivity must correspond to an increase in the aquifer resistivity, and thus a decrease in its salinity. The apparent resistivity

will also increase when the bathymetry decreases, but to a lesser extent if the aquifer is salty (Figure 2). CRP can therefore be used as a fast exploration technique to locate zones of brackish/fresh pore water. Using this technique one can easily survey multiple kilometres per day along the coastlines to detect the most vulnerable sites in terms of nutrient/contaminant leakage to the aquatic environment and loss of freshwater.

## 5 Conclusions

In this contribution, we propose an innovative combination of land ERT and marine CRP together with FDEM mapping to characterize fresh groundwater discharge. Electrical and electromagnetic methods constitute practical tools in the investigation of FSGD in coastal environments. The very high spatial lateral resolution of FDEM combined with the vertical resolution of ERT allows to interpret the presence of fresher water in 3D, while minimizing the field effort and the acquisition time. By continuing ERT profiles seaward using CRP, the surveyed area can be extended offshore to identify discharge zones located

below the low water line, even in rough areas characterized by large tidal ranges. While resistivity tomography is used to obtain a general 2D or pseudo-3D model of the salt and freshwater distribution on land, in the intertidal zone, and offshore, FDEM mapping provides information on lateral variations. Since the data acquisition is rapid and easy-to-use, this is perfectly suited for the investigation of the intertidal zone and to fill the gap between ERT profiles that take longer to acquire.

As standard output of commercially available (LIN) FDEM instrumentation systematically underestimates ECa values in

highly conductive environments, the technique has, to our knowledge, only rarely been used in the intertidal zone. However, limitations imposed by high conductive environments can be overcome through more robust interpretation of the collected data (e.g. Hanssens et al., 2019). This further extends the potential of FDEM to characterize FSGD and SI in littoral zones.

We demonstrate the ability of the proposed methodology to characterize freshwater discharge occurring at 'De Westhoek' in the western Belgian coastal plain. In this area, FSGD is present both below sea level and in the intertidal zone. Land ERT

profiles clearly identify the saltwater lens, underlain by freshwater, and originating from the infiltration of seawater on the low slope shore and discharging on the lower beach. CRP surveys further image the presence of brackish pore water and FSGD offshore. FDEM mapping in the intertidal zone allows to characterize lateral variations in the discharge zone and to locate where it becomes submarine.

To our knowledge, this constitutes the first comprehensive imaging of both the saltwater lens under the beach and the shifting

of FSGD seawards using geophysical techniques. The discharge is best-visible during low tide, since the fresh groundwater flux is higher compared to high tide. Low-tide conditions also allow to maximize the resolution of both land and marine ERT, while enabling the acquisition of FDEM data in the intertidal zone. High tide data remain necessary to ensure some overlap

with land ERT. The groundwater discharge seems to have a seasonal variability, FSGD being stronger at the end of Winter compared to the beginning of Autumn, since a warm and dry Summer precedes the latter. The comparison of this new data set to borehole logs from the 80s shows that the decrease in groundwater pumping in the dunes has strengthened the freshwater outflow in the East of the area.

## 6 Acknowledgement

The field surveys were funded by the VLIZ *Brilliant Marine Research Idea grant* (2018) and the FWO Research credit (FWO1505219N). We want to thank the VLIZ for their logistical support in the marine surveys and Liège University for lending the land ERT equipment. The authors would also like to acknowledge Josue Chishugi, Nicolas Compaire, Tim Deckmyn, Anja Derycke, Gaël Dumont, Hadrien Michel, Melissa Prieto, Mizanur Rahman Sarker, Robin Thibaut, Bart Van Impe, Valentijn Van Parys, Jan Vermaut, & Nele Vlamynck for their help with the field work. We also thank the Associate Editor Gerrit H. de Rooij and two anonymous reviewers for their comments which helped to improve this manuscript.

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

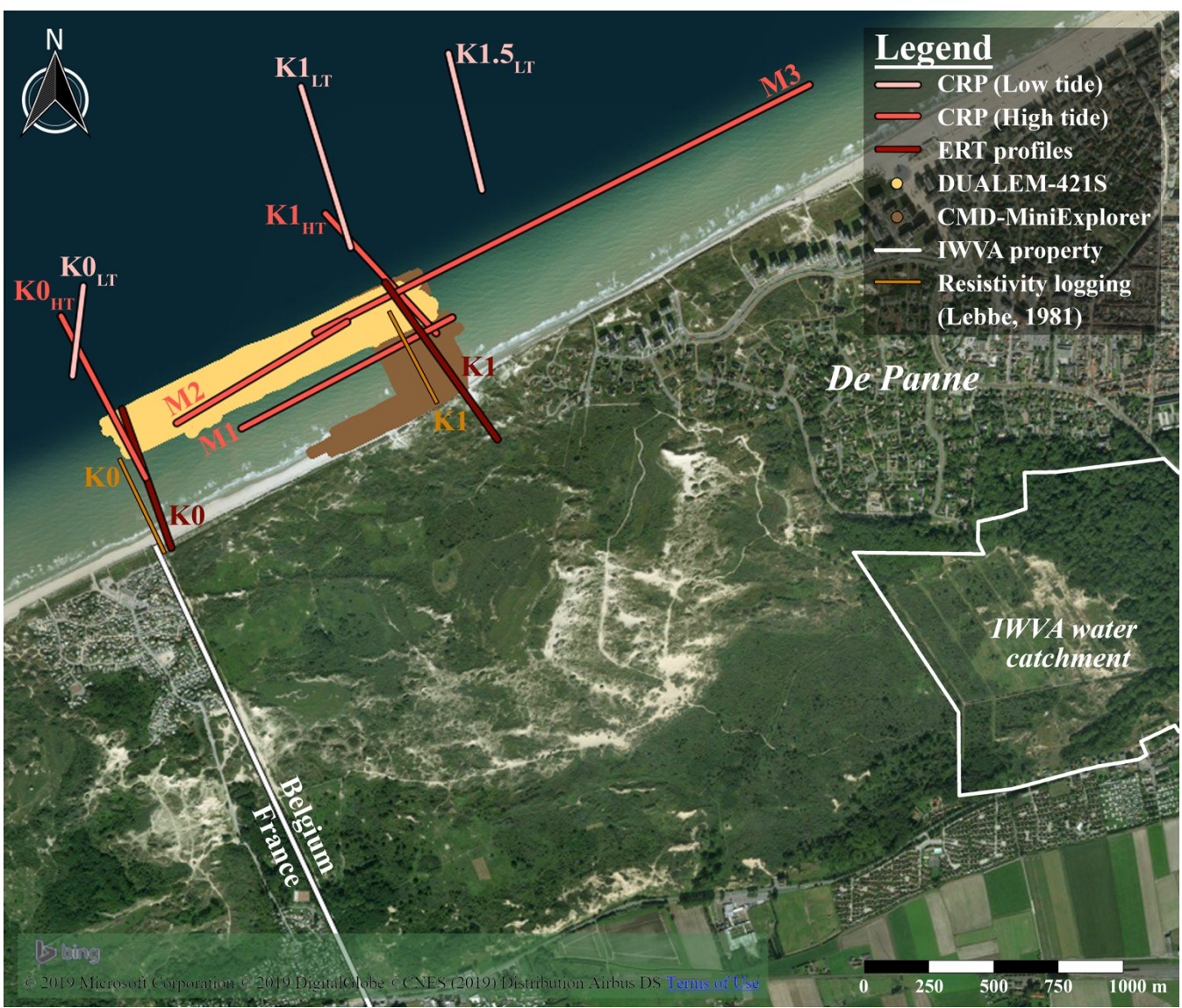

**Figure 1: The Westhoek study area, between the Belgian-French border and De Panne city. With the location of the ERT, CRP profiles, the zones of FDEM mapping, resistivity logging profiles of Lebbe (1981), and the IWVA groundwater extraction site. Copyright: Microsoft Corporation, DigitalGlobe, and CNES, 2019.**

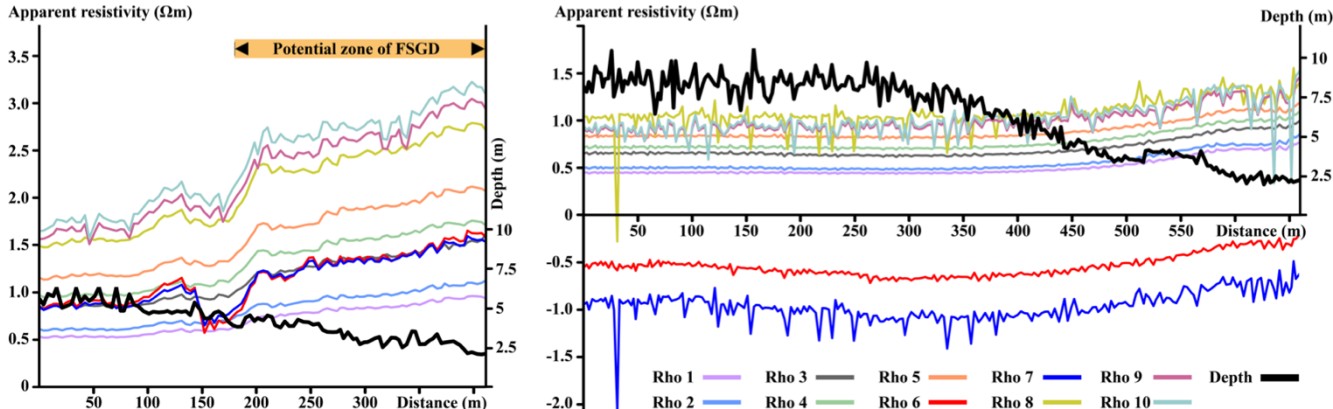

**Figure 2: Raw resistivity data plots versus bathymetry. Left: profile K0$_{HT}$: strong resistivity increase towards the beach, mainly due to SGD. Right: a perpendicular profile in front of the dunes of Wenduine (Belgium), resistivity increases only slightly with decreasing water depth.**

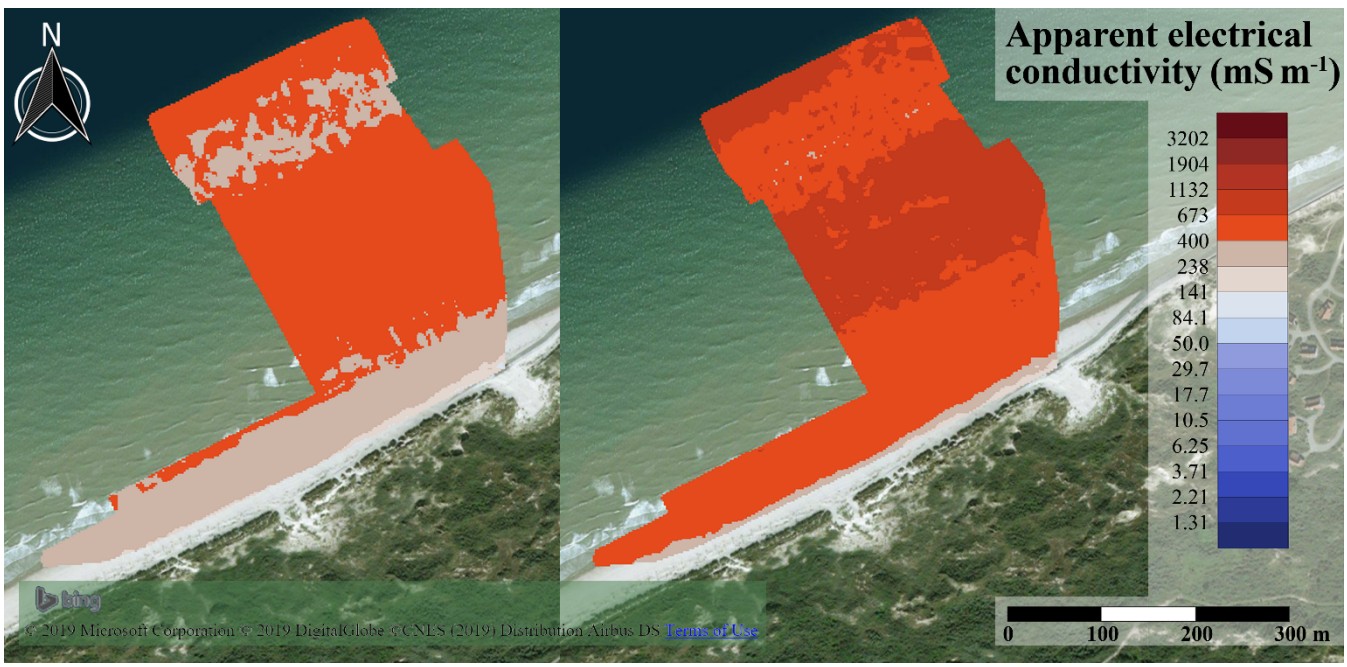

**Figure 3: The CMD-MiniExplorer ECa mapping (left) versus rECa (right), both with CMD3, pseudodepth approximately 1.8 m (23 February 2018). Copyright: Microsoft Corporation, DigitalGlobe, and CNES, 2019.**

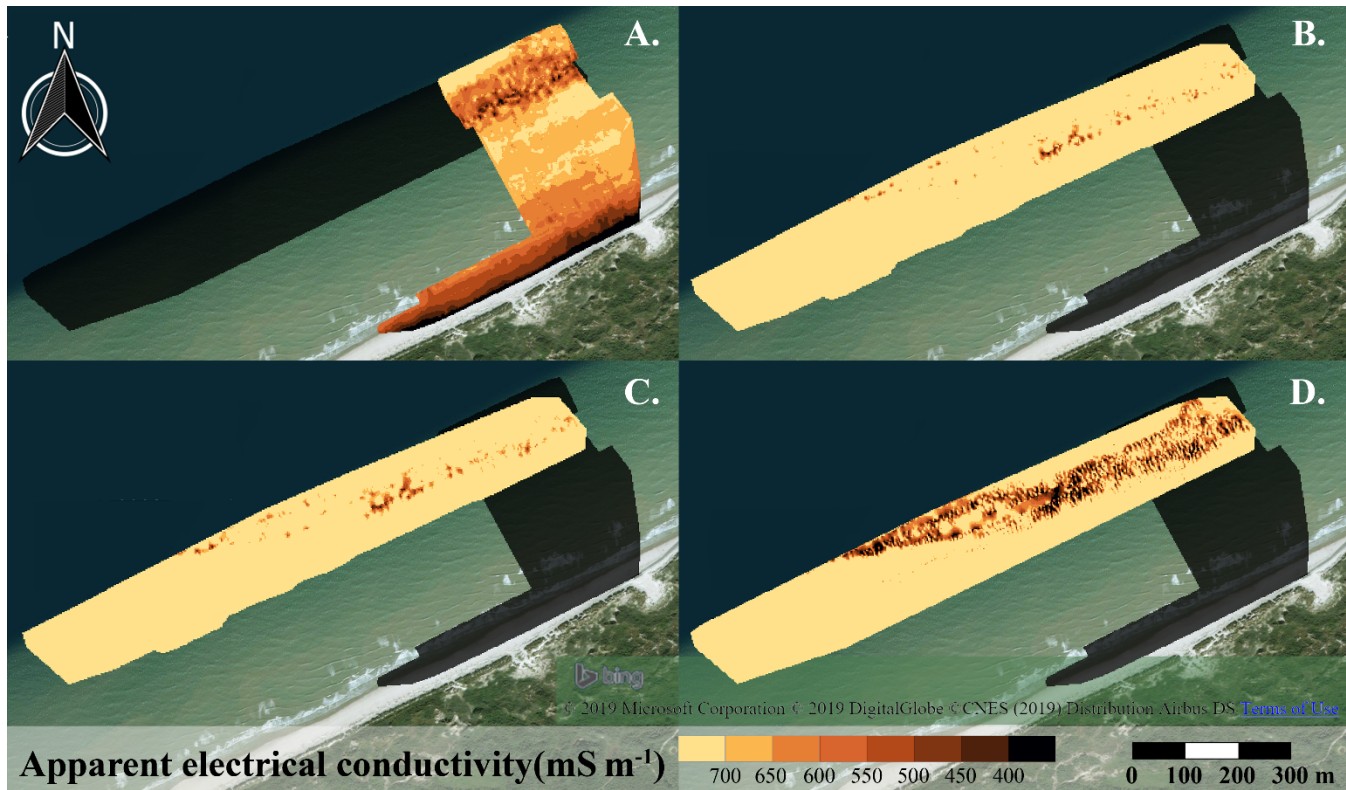

**Apparent electrical conductivity(mS m⁻¹)**

700  650  600  550  500  450  400

0   100  200  300 m

Figure 4: FDEM data. A. rECa (CMD-MiniExplorer) using CMD3, with a pseudodepth approximately 1.8 m (23 February 2018);
B., C., and D. rECa (DUALEM-421S) using respectively the HCP1, PRP4, and HCP4 configurations, providing pseudodepths of 1.5,
2, and 6 m (27-28 September 2018). Note that another colour scale is used compared to the other figures. Copyright: Microsoft
Corporation, DigitalGlobe, and CNES, 2019.

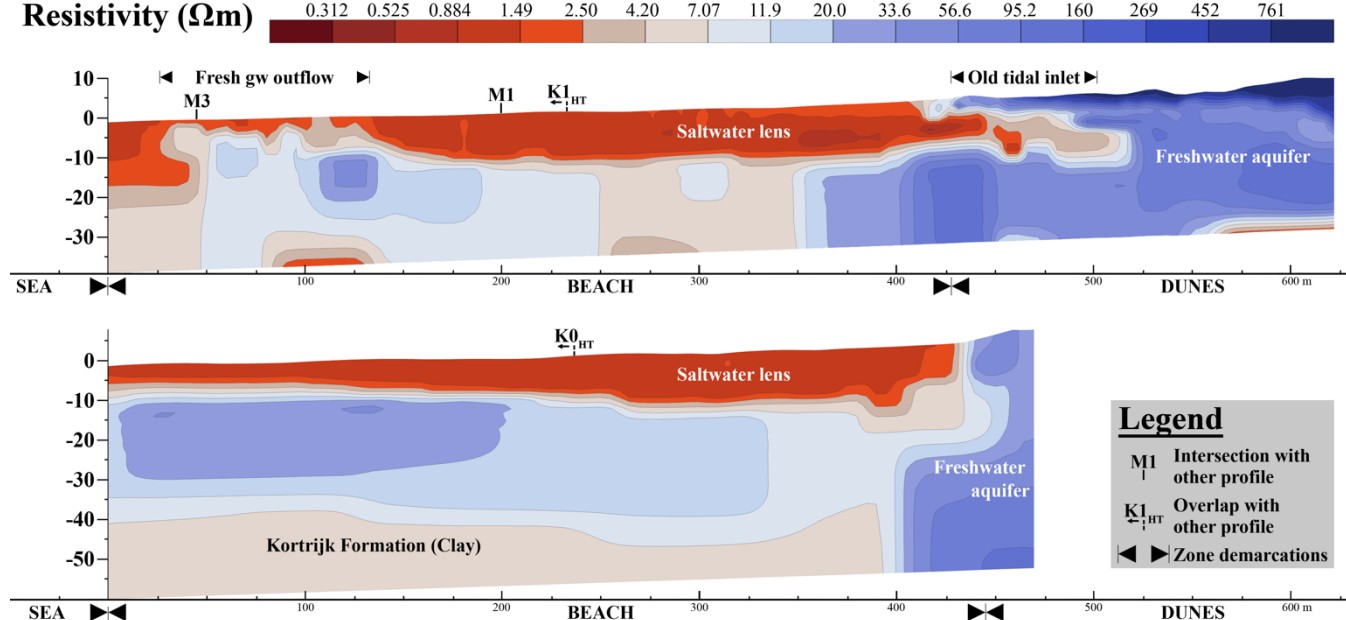

**Figure 5: ERT profiles in front of 'De Westhoek' nature reserve, perpendicular to the beach, K1 (7 March 2018) and K0 (11 October 2018). Profile numbering based on Lebbe (1981) and the water depth is in mTAW (relative to the reference level of Belgium).**

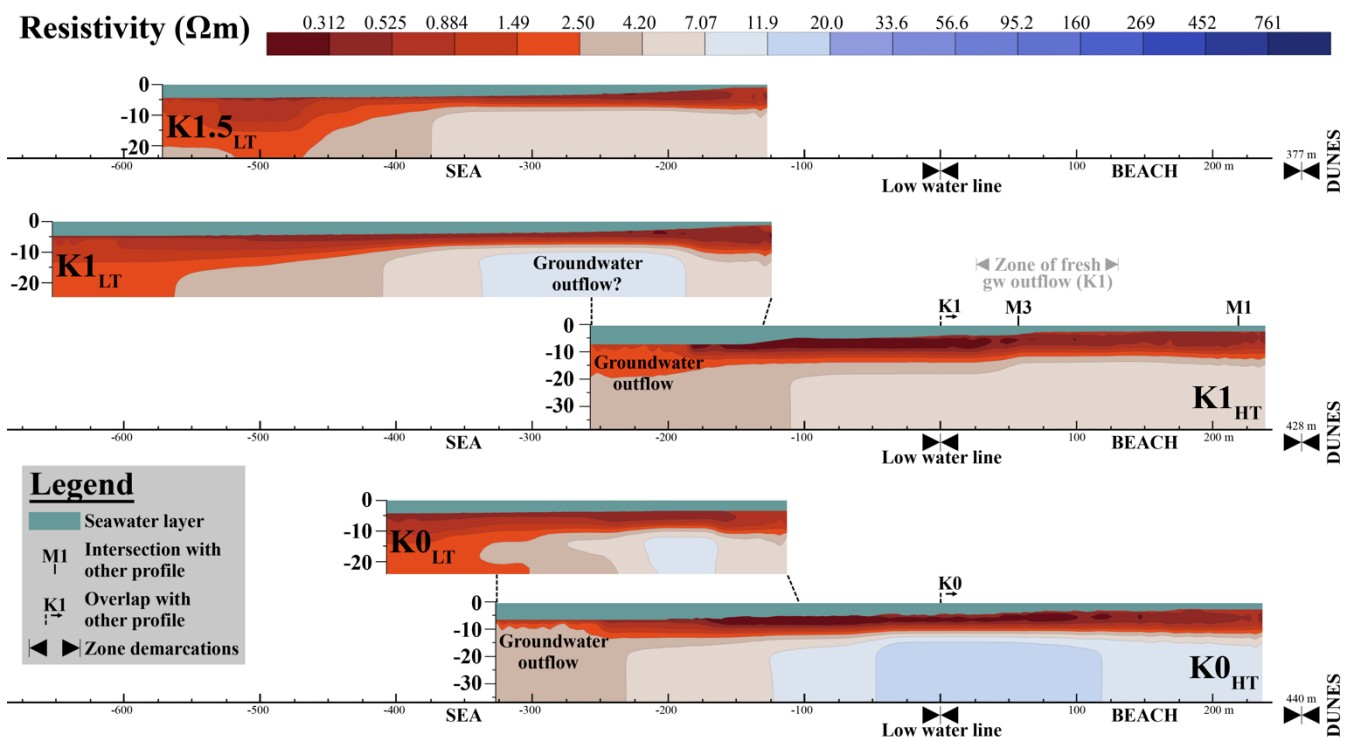

**Figure 6: CRP profiles from the Westhoek, perpendicular to the beach, acquired on 30 May 2018 (K1$_{HT}$, and K0$_{HT}$) and 22 May 2019 (K1.5$_{LT}$, K1$_{LT}$, and K0$_{LT}$). Profile numbering based on Lebbe (1981), "HT" and "LT" mean respectively high and low tide, and water depth in mTAW (relative to the reference level of Belgium).**

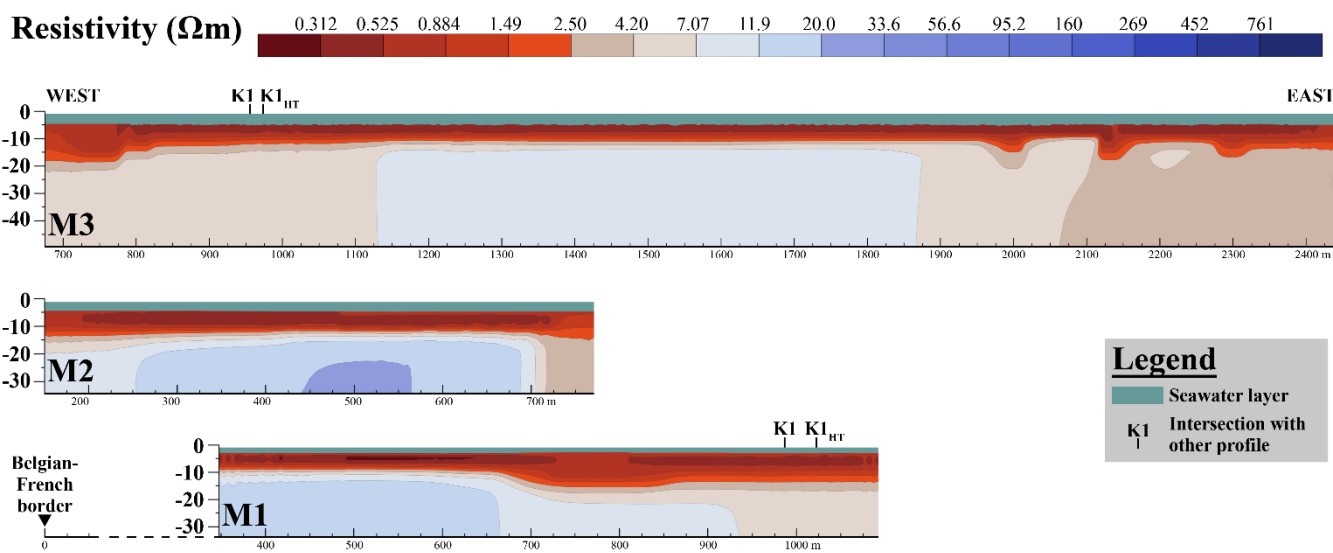

5    **Figure 7: CRP profiles parallel to the beach, taken during high tide (30 May 2018). The water depth in mTAW (relative to the reference level of Belgium).**

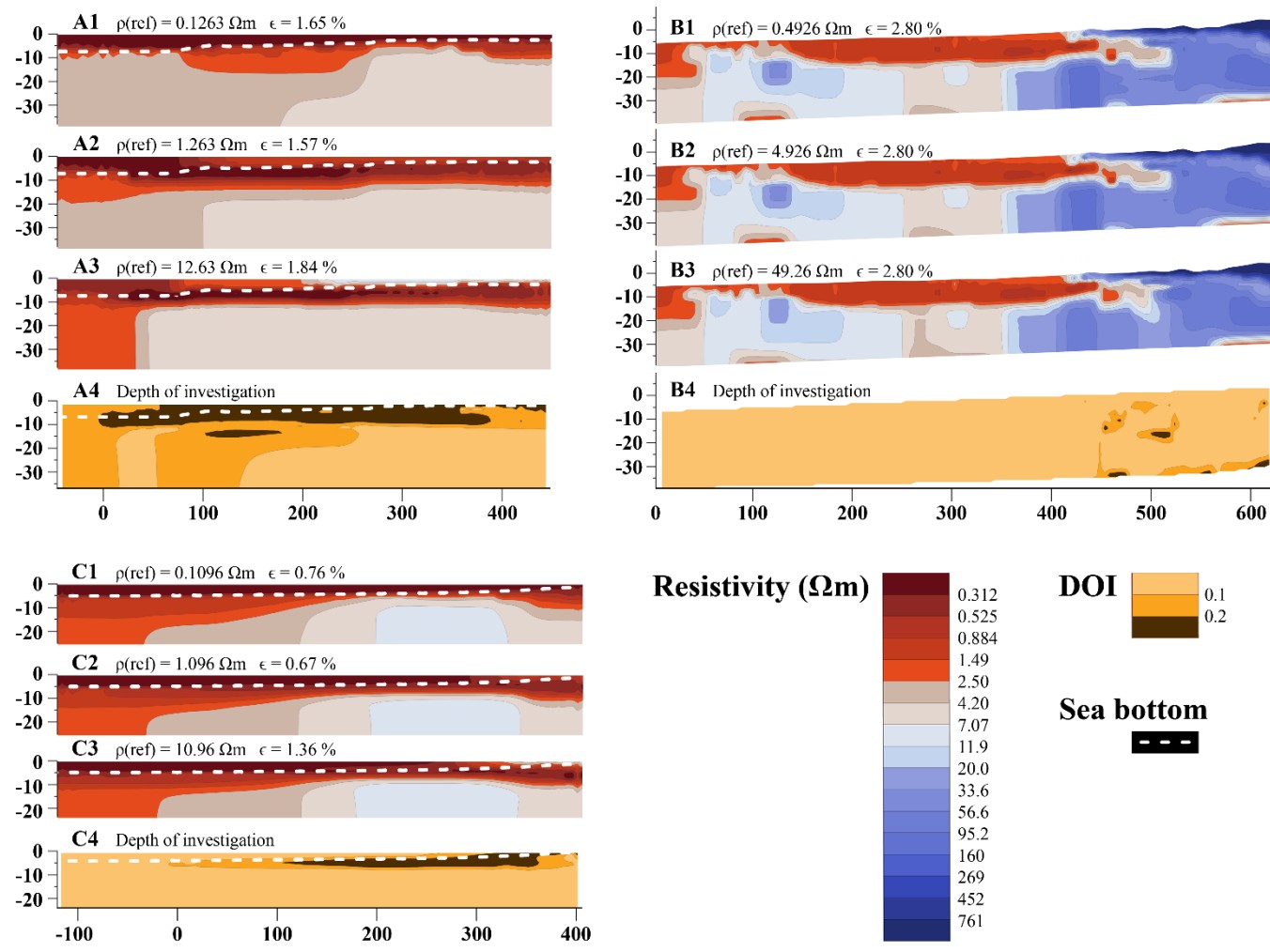

**Figure 8: Validation of the inversion models at 1 km from the Belgian-French border. A. K1$_{HT}$ marine profile; B. K1 land ERT profile; C. K1$_{LT}$ marine profile. "HT" and "LT" stand for high and low tide respectively, ρ(ref) is the reference resistivity, and ∈ the absolute error.**

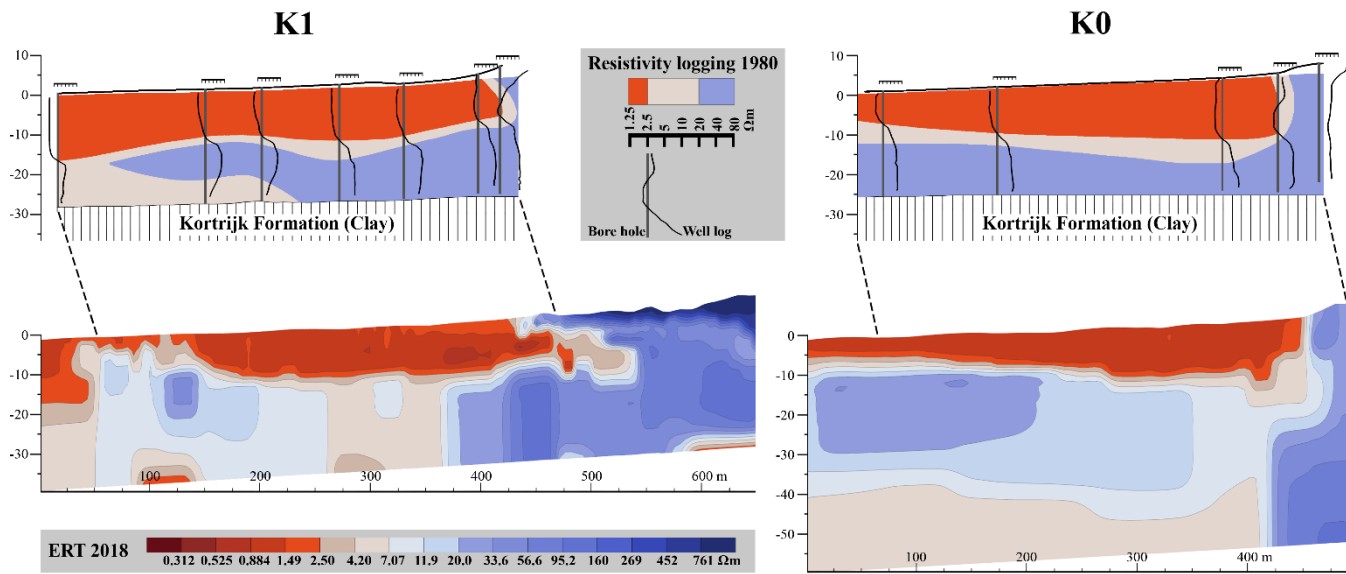

**Figure 9: Situation 1980 vs. 2018. Top: long normal resistivity logging along K1 and K0, modified from Lebbe (1981). Bottom: 2018 land ERT profiles K1 and K0.**

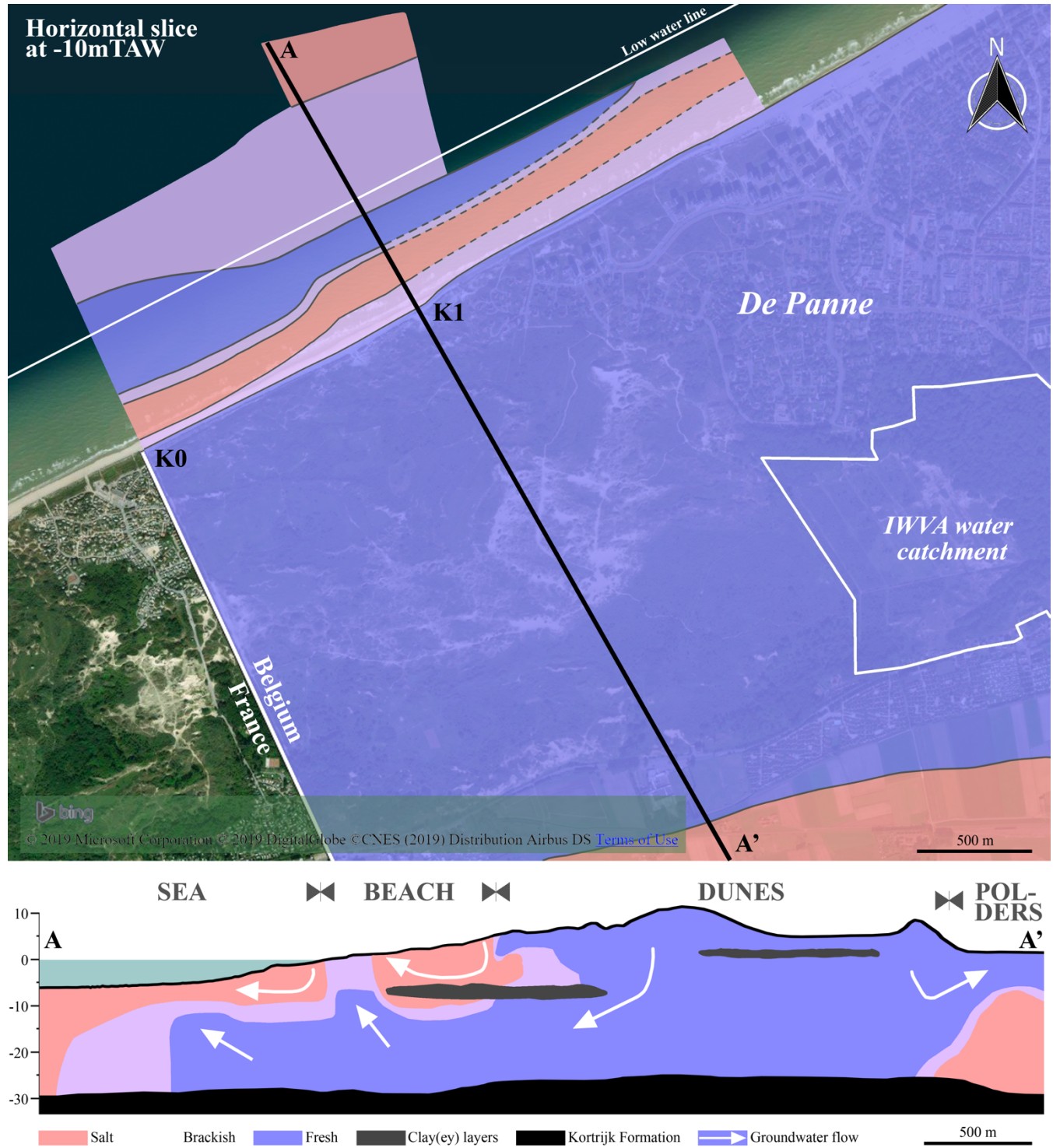

**Figure 10: Conceptual model. Based on the data presented in this study, Hermans et al. (2012), and 'de Verziltingskaart' (De Breuck et al., 1974). Copyright: Microsoft Corporation, DigitalGlobe, and CNES, 2019.**