# Peer review of "Combining resistivity and frequency domain electromagnetic methods to investigate submarine groundwater discharge (SGD) in the littoral zone"

_Hydrology and Earth System Sciences, 2019_

## Referee Comment (RC1) · Anonymous Referee #1 · 4 Feb 2020

This manuscript presents an interesting set of methods and data to generate a complete overview of a study area on the coast of Belgium. The use of geophysical methods is very extended in the study of groundwater in coastal setting and combined methods optimizing their use can be applied in other regions. There are several relevant facts presented and discussed that can be valuable for the scientific community but the text needs improvement in multiple sections to provide a more specific message and to align better the content of the manuscript.

Specific comments

Abstract

The abstract can be improved. General descriptions as lines 10-14 or very specific details about geophysical methods can be replaced by a more clear overview of the research conducted and the new insights provided by this manuscript.

Introduction

The concept of SGD has to be polished. Along with the text, it can be inferred that authors refer in most of the cases to fresh or terrestrial SGD even if in the introduction there is a brief mention about recirculated/saline SGD. I think the authors should update this way of referring to SGD. To get an update about this topic they can start by the beginning:

Moore, W.S., Church, T.M., 1996. Submarine groundwater discharge. Reply to Younger (1996). Nature 382, 122.

And keep track of the recent advances especially about the saline part of SGD that is what it is usually a more unknown element for the hydrology community:

Rodellas, V., Stieglitz, T. C., Andrisoa, A., Cook, P. G., Raimbault, P., Tamborski, J. J., Radakovitch, O. (2018). Groundwater-driven nutrient inputs to coastal lagoons: The relevance of lagoon water recirculation as a conveyor of dissolved nutrients. Science of the Total Environment, 642, 764-780

Or by recent reviews/discussions:

Taniguchi, M., Dulai, H., Burnett, K. M., Santos, I. R., Sugimoto, R., Stieglitz, T., Burnett, W. C. (2019). Submarine groundwater discharge: Updates on its measurement techniques, geophysical drivers, magnitudes, and effects. Frontiers in Environmental Science 7.

Duque, C., Michael, H.A., Wilson, A.M. (2020). The subterranean estuary: Technical term, simple analogy, or source of confusion? Water Resources Research 56.

The introduction would benefit from being more specific, in its current state it is too

broad without a clear thread about what wants to be showed. The review of multiple cases studying SGD is not really showing what are the gaps in the current use of methods or research questions that need to be addressed. For example lines 15-17, page 3 are out of context and could be deleted as well as page 2, lines 31-33.

On the contrary, page 3, lines 26-31 is a good example of how should be done the introduction, adding references to these sentences would be a better way to probe the utility of this study showing cases about SGD with spatial and temporal variability/ the challenge of work in coastal settings /studies discussing the problem of the gap between land and sea. I suggest a full reorganization of this section rewriting part of it.

Some additional references that can be useful:

Stieglitz, T., Rapaglia, J., Bokuniewicz, H. (2008). Estimation of submarine groundwater discharge from bulk ground electrical conductivity measurements. J. Geophys. Res. Ocean. 113, 1–15.

Stieglitz, T., Taniguchi, M., Neylon, S. (2008). Spatial variability of submarine groundwater discharge, Ubatuba, Brazil. Estuar. Coast. Shelf Sci. 76, 493–500.

Kinnear, J.A., Binley, A., Duque, C., Engesgaard, P.K. (2013). Using geophysics to map areas of potential groundwater discharge into Ringkøbing Fjord, Denmark. Lead. Edge 32.

Study area

The geology elements used in the discussion must be initially presented in the study area section. For example, it is mentioned a clay layer very important for the interpretation of the data but it is not explained previously. Also, this layer can be probably inserted in the geophysical interpretation in the figures as lithological columns (or any other graphical way) to probe the reliability of the method.

What is the local water problem that wants to be solved in the study area? It is mentioned in the text but not clearly. Is any interest in increasing the pumping rate? In the

text, it is said that the pumping rate has been decreased in the last years.

Methodology

In general, it should be mentioned how many measurements/cross-sections have been collected for each method.

Page 5, line 20. Is this needed to be explained for any reason? If the collection is easier and faster there is no reason to use the other method.

Page 5, lines 24-25. Is this an innovation of this study compared with others?

Page 6 lines 10-12. I think the data generated by broken electrodes must be removed both from the text and the figures, they do not provide any value.

Results and discussion

One of the main issues in the text is the disorganization in the section results and discussion. This makes it difficult to read and the final message is not well transmitted. I strongly encourage a complete reorganization of the results and discussion, for example by dividing the text into the following sections with a heading:

- CRP data (clear specific presentation of the data without mixing with other sections)

- FDEM data (clear specific presentation of the data without mixing with other sections)

- ERT data (clear specific presentation of the data without mixing with other sections)

- Advantages of the combined methodology (compared with previous studies)

- Geophysical innovations (technical improvements, novelties, and new approaches)

- Seasonal SGD changes and over the last 20 years (showing the data from previous reports)

I think that the approach to present the data as "they were collected in the field" is not working well enough to be justified.

Along with the text, there are multiple mentions to a "freshwater tongue" below the salty water. This is a well-established field of research that the authors should check to put into context their findings.

For example:

Robinson, C., Gibbes, B., Li, L. (2006). Driving mechanisms for groundwater flow and salt transport in a subterranean estuary. Geophys. Res. Lett. 33, 3–6. . . .And all the following papers based on modeling and field observations. There are dozens but this is one of the first ones.

Avoid referring to figures as right/left side, better say any cardinal direction or even mark in the figure the parts that want to be highlighted.

Page 9, line 1. How do you know the origin of the brackish water?

Page 9, lines 8-12. The discussion about the comparison with previous surveys is totally out of context, as said before, better create a full section of comparison where the old data are also presented maybe even graphically. This applies also for the following comparisons in other pages.

Page 10, lines 10-13. This paragraph seems to be out of context. Better move a section about geophysical innovations/progress as said before. The same for other paragraphs where geophysical technical aspects are commented.

Page 10, line 22. What is the dike between the beach and the dunes? This should be presented in the study area section.

Page 10, line 22. The effect of local heterogeneities is ambiguous, specify better or remove.

Page 11. Line 2, is an example of the wrong use of recirculated water. The authors can refer simply to mixing between fresh and salty, or maybe to variable density-driven flow, but recirculated here is not meaning what the authors want. In general. they should

check this along with all the text after reading the general references provided before.

Page 11, lines 10-20. These methods of testing the results should more clearly be specified in the methods section.

Conclusions

Following the structure proposed for results and discussion, the conclusions can be more structured highlighting the novelties and new insights of this study. Page 12, line 15-16. Not needed, delete.

Figures

The figures are especially relevant in this manuscript as they contain all the data presented, it is, therefore, essential that they are as clear as possible.

The figure captions can be improved, technical data as error and dates can be removed (these data can be added in a table for example or commented in the text). A legend including all the lines in the figures (dotted black, dotted white, continuous...) would help in the interpretation, it is quite difficult for the reader to do it in its current state.

Another important improvement would be to differentiate the water column over the aquifer. As this is a research about groundwater, the properties of the sea are not that needed in the figures and would help in the graphical interpretation of the results.

If there are specific areas of the cross-sections the authors want to refer to during the explanations, they can mark them in the figures to facilitate the link between text and figures.

I understand that the location of the cross-section in the figures corresponds with the spatial location, even if this is not clearly stated. The authors can add a reference point so the reader can know always how far is from the shoreline (i.e, distance to the high/low tide mark/dunes/).

Also, all the profiles must have the same markers, for example, the dash lines in figure

4 are only in two of the cross-sections. In general, any improvement to give a better overview of the location of the cross-section would benefit the figures.

It is not clear reading the figure caption of figure 7 what are the differences between A, B, C and D. Figure 8 is difficult to follow even after reading the text (maybe because it is quite brief). A presentation of the results adapted to the purpose might be considered instead of presenting the data in the same format as in the analysis of the results (for example showing the difference in resistivity between inversion models).

---

## Referee Comment (RC2) · Anonymous Referee #2 · 12 Feb 2020

Review Manuscript Number: hess-2019-540 Title: Combining resistivity and frequency domain electromagnetic methods to investigate submarine groundwater discharge (SGD) in the littoral zone Article Type: Research paper

The manuscript by Paepen,et al. addresses an important issue when it comes to SGD: the view across compartments. Traditional work mostly focuses either on marine or terrestrial compartments. Only a few studies, such as the present one, consider both compartments and see SGD as a continuum and across-compartment process. As such it is of utmost importance to include geophysical approaches that to date are still underrepresented despite its promising results (see works of Swarzenski et al., Cross et al., Virtasalo et al.). Despite the importance the manuscript contains drawbacks both in the way it is presented and concerning content-wise aspects. I addressed both in the attached *.pdf and partly in the questionnaire, I was asked to answer (see below).

Questionnaire:

Scientific significance: Does the manuscript represent a substantial contribution to scientific progress within the scope of Hydrology and Earth System Sciences (substantial new concepts, ideas, methods, or data)? Fair

Scientific quality: Are the scientific approach and applied methods valid? Are the results discussed in an appropriate and balanced way (consideration of related work, including appropriate references)? Fair

Presentation quality: Are the scientific results and conclusions presented in a clear, concise, and well-structured way (number and quality of figures/tables, appropriate use of English language)? Fair

1. Does the paper address relevant scientific questions within the scope of HESS? The manuscript by Paepen et al. addresses an important issue when it comes to SGD, the view across compartments. Traditional work mostly focuses either on marine or terrestrial compartments. Only a few studies, such as the present one, consider both compartments and see SGD as a continuum and across-compartment process. As such it is of utmost importance to include geophysical approaches that to date are still underrepresented despite its promising results (see works of Swarzenski et al., Cross et al., Virtasalo et al.). Concluding, yes it addresses a very relevant scientific question.

2. Does the paper present novel concepts, ideas, tools, or data? According to the authors, only the combination of the three applied geophysical methods allow to "to fully characterize the discharge zone in the target area (Pg4 L4)". The authors mention the combination of methods to be used for the first time in the context of SGD, which makes it novel.

3. Are substantial conclusions reached? This question is difficult to answer from my perspective. Presented results show the same pattern of either resistivity or conductivity signals. Following the authors narrative and conclusion it seems as if substantial conclusion are reached. Yet, as a non-geophysicist it is very often difficult to follow the narrative as the authors refer to likeliness of fresh/brackish discharge but show resistivity or conductivity values. Besides, results are obtained without validating them. Even the inversion was based on geophysical data only although the authors themselves state "Due to measurement errors, resolution and inversion constraints it is impossible to deduce the true total dissolved solids (TDS) without a specific calibration of geophysical measurements based on ground truth data. (Pg8 L10f)". The statement is addressed to the deduction of TDS but given the measurement uncertainties and natural heterogeneities along with small variances between brackish, fresh water along with its effects on resistivity or conductivity signals, the above question cannot be properly answered. However, I must emphasize, this is from the perspective of a non-geophysicist. I certainly feel not qualified to evaluate the methods in general and leave this part to more qualified colleagues. Instead, I concentrated on the general picture and the benefit for SGD investigations.

4. Are the scientific methods and assumptions valid and clearly outlined? As written before, I am not qualified to properly answer the question concerning the methodical part. In parts, the paper reads too technical and is hard to understand. I added comments in the *.pdf. From my limited understanding, and also the authors themselves mention it several times throughout the manuscript, I would like to see the pros/cons of each method (vertical resolution, max. vertical range, uncertainties etc.) to be in the position to better judge.

5. Are the results sufficient to support the interpretations and conclusions? The combination of the three methods, as always when it comes to combine different methods in order to reduce uncertainties of single methods, the interpretations are supported by the results. Certain conclusion on fluxes should be avoided (Pg10 L16 // Pg 12 L8).

This would mean to include a time-dimension which the mono-temporal data do not allow. Moreover, what would certainly improve the narrative and alleviate the understanding would be figures that i) include a scale showing the likeliness of SGD, ii) have information in the graphics concerning important features (salt water lens, freshwater, clay lens etc.) and iii) even a continuous spatial scale beneath Figs. 4,5,6 to be able to follow the distances given in the text. I wished the authors had added some in-situ data on e.g. pore water chemistry, drilling etc. to prove the results with an independent data set which would elevate the manuscript to a higher level.

6. Is the description of experiments and calculations sufficiently complete and precise to allow their reproduction by fellow scientists (traceability of results)? I would leave this question to colleagues with a profound understanding/knowledge of the methods. Used software was mentioned, likewise were applied thresholds.

7. Do the authors give proper credit to related work and clearly indicate their own new/original contribution? Yes.

8. Does the title clearly reflect the contents of the paper? Yes.

9. Does the abstract provide a concise and complete summary? Yes and no. Complete yes, concise partly. Parts of the abstract between L18-24 could be reduced/deleted as it unnecessarily lengthens the abstract. I leave it the authors to decide.

10. Is the overall presentation well structured and clear? Yes. However, I would encourage the authors to add a subsection in which they outline which of the results can be stated with certainty given all uncertainties and relative terms/subjectivity, and which of the results must be proven as they are highly uncertain. I added comments at the respective parts of the manuscript in the *.pdf

11. Is the language fluent and precise? Yes, it is well written.

12. Are mathematical formulae, symbols, abbreviations, and units correctly defined and used? Not applicable.

13. Should any parts of the paper (text, formulae, figures, tables) be clarified, reduced, combined, or eliminated? Yes, I added suggestions in the *.pdf

14. Are the number and quality of references appropriate? Yes.

15. Is the amount and quality of supplementary material appropriate? Not applicable.

Please also note the supplement to this comment:
https://www.hydrol-earth-syst-sci-discuss.net/hess-2019-540/hess-2019-540-RC2-supplement.pdf

**Supplement:**

[revised manuscript text omitted]

---

## Author Comment (AC1) · 30 Mar 2020

This manuscript presents an interesting set of methods and data to generate a complete overview of a study area on the coast of Belgium. The use of geophysical methods is very extended in the study of groundwater in coastal setting and combined methods optimizing their use can be applied in other regions. There are several relevant facts presented and discussed that can be valuable for the scientific community but the text needs improvement in multiple sections to provide a more specific message and to align better the content of the manuscript.

» We would like to thank anonymous referee #1 for the very constructive review. It is

clear that part of the message we wanted to deliver is lost by the way the manuscript is structured, especially the results section. Below we give an answer to all the specific comments (our responses are between: "» «"). The necessary changes will be made in the revised manuscript. «

ABSTRACT

The abstract can be improved. General descriptions as lines 10-14 or very specific details about geophysical methods can be replaced by a more clear overview of the research conducted and the new insights provided by this manuscript.

» We believe that a good abstract should always first describe the general context. Before highlighting the techniques we have used and the complementary methodology, it is thus important to address the importance of submarine groundwater discharge. This indicates why it is important to study this phenomenon. We will remove specific details, such as page 1 lines 18-22. And insist on the new insights brought by our methodology. «

INTRODUCTION

The concept of SGD has to be polished. Along with the text, it can be inferred that authors refer in most of the cases to fresh or terrestrial SGD even if in the introduction there is a brief mention about recirculated/saline SGD. I think the authors should update this way of referring to SGD. To get an update about this topic they can start by the beginning: Moore, W.S., Church, T.M., 1996. Submarine groundwater discharge. Reply to Younger (1996). Nature 382, 122. And keep track of the recent advances especially about the saline part of SGD that is what it is usually a more unknown element for the hydrology community: Rodellas, V., Stieglitz, T. C., Andrisoa, A., Cook, P. G., Raimbault, P., Tamborski, J. J., Radakovitch, O. (2018). Groundwater-driven nutrient inputs to coastal lagoons: The relevance of lagoon water recirculation as a conveyor of dissolved nutrients. Science of the Total Environment, 642, 764-780

» We indeed refer in most cases to only terrestrial/fresh SGD. In particular, using resistivity methods, we rely on the contrast of salinity to identify fresh/terrestrial water discharge. Detecting saline SGD is not possible with the used methods. We anticipated that most readers would be familiar with this broader context so that we focused on geophysics in the introduction. Since this is unclear, we will make this clearer throughout the text and keep the suggested references in mind. «

Or by recent reviews/discussions: Taniguchi, M., Dulai, H., Burnett, K. M., Santos, I. R., Sugimoto, R., Stieglitz, T., Burnett, W. C. (2019). Submarine groundwater discharge: Updates on its measurement techniques, geophysical drivers, magnitudes, and effects. Frontiers in Environmental Science 7. Duque, C., Michael, H.A., Wilson, A.M. (2020). The subterranean estuary: Technical term, simple analogy, or source of confusion? Water Resources Research 56.

» Thank you for highlighting these interesting, recent papers. We will incorporate them in the introduction. «

The introduction would benefit from being more specific, in its current state it is too broad without a clear thread about what wants to be showed. The review of multiple cases studying SGD is not really showing what are the gaps in the current use of methods or research questions that need to be addressed. For example lines 15-17, page 3 are out of context and could be deleted as well as page 2, lines 31-33. On the contrary, page 3, lines 26-31 is a good example of how should be done the introduction, adding references to these sentences would be a better way to probe the utility of this study showing cases about SGD with spatial and temporal variability/the challenge of work in coastal settings /studies discussing the problem of the gap between land and sea. I suggest a full reorganization of this section rewriting part of it.

» In the introduction, we try to focus on the use of geophysics in coastal environments, because the methodology we propose is based on such techniques. It might indeed seem very or even too broad. It gives an overview of the most common uses of geophysical methods in a coastal zone, their potential and their limitations. We will focus on the most important aspects (lateral and vertical resolution, acquistition speed) and put more emphasis on the gaps and addition this new methodology brings. This was the main objective of the paragraph page 3, L26-31. «

Some additional references that can be useful: Stieglitz, T., Rapaglia, J., Bokuniewicz, H. (2008). Estimation of submarine groundwater discharge from bulk ground electrical conductivity measurements. J. Geophys. Res. Ocean. 113, 1–15; Stieglitz, T., Taniguchi, M., Neylon, S. (2008). Spatial variability of submarine groundwater discharge, Ubatuba, Brazil. Estuar. Coast. Shelf Sci. 76, 493–500. Kinnear, J.A., Binley, A., Duque, C., Engesgaard, P.K. (2013). Using geophysics to map areas of potential groundwater discharge into Ringkøbing Fjord, Denmark. Lead. Edge 32.

» Thank you for pointing these out. «

STUDY AREA

The geology elements used in the discussion must be initially presented in the study area section. For example, it is mentioned a clay layer very important for the interpretation of the data but it is not explained previously. Also, this layer can be probably inserted in the geophysical interpretation in the figures as lithological columns (or any other graphical way) to probe the reliability of the method.

» We are referring to the clay layer which bounds the phreatic aquifer in the text. This is the Kortrijk Formation, which is discussed in this section. In the results section, we also mention a local clay layer (page 9 line 3) in the tidal inlet. A description of this one will be included in the study area. «

What is the local water problem that wants to be solved in the study area? It is mentioned in the text but not clearly. Is any interest in increasing the pumping rate? In the text, it is said that the pumping rate has been decreased in the last years.

» The pumping rate has indeed been decreased, so at the moment there is - at that

location - no problem with attraction of salt water from the sea. But the potable water is scarce in the region, especially in Summer when lots of tourists visit, and so a comprehensive understanding of the hydrogeological system is important for the sustainable management of the "shallow" groundwater reserves, this will be explained in the study area section. By comparing with Lebbe (1981), we can see how the decrease in pumping affects the freshwater discharge. And the idea is to compare this study area, in a next step, with another zone of the Belgian coast where larger quantities (around 10 times more) are being pumped from the dune aquifer, but where they also infiltrate treated wastewater to enable this. But it is, at the moment, not possible to increase the pumping rate at the Westhoek mainly for two reasons. First, the facility is located in a nature reserve and so there are strict regulations. Also, some wells show an increased salinity, since old seawater from the polders (which is much closer to the wells compared to the sea) is being attracted. «

METHODOLOGY

In general, it should be mentioned how many measurements/cross-sections have been collected for each method.

» To complement the information in the map (Figure 1), we will include the details it in the methodology section. «

Page 5, line 20. Is this needed to be explained for any reason? If the collection is easier and faster there is no reason to use the other method.

» With continuous resistivity profiling (CRP), a larger part of the injected current directly flows in the conductive saltwater layer. And so, the sensitivity of marine electrical resistivity (MER) to resistivity variations in the sediments will be higher, so there are cases where you could opt for MER. This is already described in the introduction, page 3 lines 10-16. «

Page 5, lines 24-25. Is this an innovation of this study compared with others?

» Most other studies are performed in zones that have smaller tidal differences and which are less rough compared to the North Sea. In this study, we prove that it is possible to perform these kind of measurements on the North Sea or in similar conditions. «

Page 6 lines 10-12. I think the data generated by broken electrodes must be removed both from the text and the figures, they do not provide any value.

» We are open on the set-backs we had with the equipment. It shows that with less channels, you can still obtain reliable inversion models. It also has an impact on the sensitivity of the high tide inversion models compared to the ones from low tide conditions (less data are available). «

RESULTS AND DISCUSSION

One of the main issues in the text is the disorganization in the section results and discussion. This makes it difficult to read and the final message is not well transmitted. I strongly encourage a complete reorganization of the results and discussion, for example by dividing the text into the following sections with a heading:

- CRP data (clear specific presentation of the data without mixing with other sections)

- FDEM data (clear specific presentation of the data without mixing with other sections)

- ERT data (clear specific presentation of the data without mixing with other sections)

- Advantages of the combined methodology (compared with previous studies)

- Geophysical innovations (technical improvements, novelties, and new approaches)

- Seasonal SGD changes and over the last 20 years (showing the data from previous reports)

I think that the approach to present the data as "they were collected in the field" is not working well enough to be justified.

» We opted for the "field collection order", because it justified the choices we made during the 2 years of the field campaign, starting with the Eastern part of the study area, and then moving to the West. We understand this might be more difficult to read, since we mix multiple methods and figures in many of the paragraphs. We will therefore follow your recommendation in the revised manuscript. Based on the above proposed sections, we will create the following subdivisions: 1) FDEM data, since this is the first method we explored. 2) Land ERT data, which proves that the features observed with FDEM are in fact related to freshwater discharge. 3) CRP data, to extend the ERT profiles, allowing to visualize groundwater discharge offshore. 4) Quality appraisal of the ERT/CRP inversion models, explaining figure 8 in detail. 5) Seasonal variation of the discharge. 6) Long-term evolution, comparing with data from 1980. 7) Advantages of the proposed methodology. «

Along with the text, there are multiple mentions to a "freshwater tongue" below the salty water. This is a well-established field of research that the authors should check to put into context their findings. For example: Robinson, C., Gibbes, B., Li, L. (2006). Driving mechanisms for groundwater flow and salt transport in a subterranean estuary. Geophys. Res. Lett. 33, 3–6. : : :And all the following papers based on modeling and field observations. There are dozens but this is one of the first ones.

» We feel that this does not fall in the scope of this study. The objective of this study is not to describe this phenomenon as a new discovery (this is indeed well documented), but to fully image this saltwater lens using geophysics. How this lens is formed in the Westhoek was already extensively discussed by Vandenbohede, A. and Lebbe, L.: Occurrence of salt water above fresh water in dynamic equilibrium in a coastal groundwater flow system near De Panne, Belgium, Hydrogeological Journal, 14(4), 462-472, https://doi.org/10.1007/s10040-005-0446-5, 2006a. We think this reference is the most appropriate as it describes the situation at the study site. In fact, the western Belgian coastal plain is one of the first area where this freshwater tongue was observed, by Luc Lebbe and his co-authors back in the 70s and 80s. At that time,

there was serious doubts about the possible occurrence of saltwater above freshwater. «

Avoid referring to figures as right/left side, better say any cardinal direction or even mark in the figure the parts that want to be highlighted.

» This might indeed be confusing. We'll change this to North, East, South, West or dune-/seaside. «

Page 9, line 1. How do you know the origin of the brackish water?

» The artificial tidal inlet was monitored very closely during multiple years with EM borehole logging, since it was potentially threatening the freshwater reserves, this will be more clearly indicated in the text. Those data clearly shows an initially freshwater aquifer progressively infiltrated by saltwater. Now that infiltration does not occur anymore, the hydraulic gradient tends to transport this water back to the sea. Recent EM logs show that the salinity progressively decreases. We, therefore, know that seawater entered the inlet and a remnant of that is still present in the aquifer. «

Page 9, lines 8-12. The discussion about the comparison with previous surveys is totally out of context, as said before, better create a full section of comparison where the old data are also presented maybe even graphically. This applies also for the following comparisons in other pages.

» See previous comment on the structure, we agree that this might be easiest to read. An extra figure will be included to graphically show the differences. «

Page 10, lines 10-13. This paragraph seems to be out of context. Better move a section about geophysical innovations/progress as said before. The same for other paragraphs where geophysical technical aspects are commented.

» We agree. We will include this comment in the section dedicated to the geophysical innovation. «

Page 10, line 22. What is the dike between the beach and the dunes? This should be presented in the study area section.

» It should be in fact "dyke", it is an embankment constructed in front of the coastal cities, to prevent from flooding. «

Page 10, line 22. The effect of local heterogeneities is ambiguous, specify better or remove.

» There are clay lenses within the phreatic aquifer, and we think those might have an influence on the local distribution of fresh and salt water. However, clays have a low resistivity and cannot be easily identified in saline environment. We will extend the explanation. «

Page 11. Line 2, is an example of the wrong use of recirculated water. The authors can refer simply to mixing between fresh and salty, or maybe to variable density-driven flow, but recirculated here is not meaning what the authors want. In general, they should check this along with all the text after reading the general references provided before.

» As mentioned before, we will make a better distinction between terrestrial FSGD and a mix of saline and fresh discharge. «

Page 11, lines 10-20. These methods of testing the results should more clearly be specified in the methods section.

» The calculation of the DOI is a common way to assess the robustness of ERT inversion results. There is a large paragraph in the processing section (3.1.3), this addresses the validation method that is used, this part will be expanded into a separate section (3.1.4. Interpretation and Inversion model appraisal). For additional information on the DOI, we refer to Oldenburg, D.W., and Li, Y. 1999 [Estimating depth of investigation in dc resistivity and IP surveys, Geophysics, 64(2), 403-416, https://doi.org/10.1190/1.1444545] and others. We think that the new structure for the result section will also help to present this validation more clearly. «

CONCLUSIONS

Following the structure proposed for results and discussion, the conclusions can be more structured highlighting the novelties and new insights of this study.

» The novelties will be better highlighted. «

Page 12, line 15-16. Not needed, delete.

» This sentence will be removed. «

FIGURES

The figures are especially relevant in this manuscript as they contain all the data presented, it is, therefore, essential that they are as clear as possible. The figure captions can be improved, technical data as error and dates can be removed (these data can be added in a table for example or commented in the text). A legend including all the lines in the figures (dotted black, dotted white, continuous: : :) would help in the interpretation, it is quite difficult for the reader to do it in its current state.

» We will include a legend in the figures which contain ERT profiles. The full black lines are at intersections between resistivity profiles, dotted black lines show the overlap between profiles, and the dotted white line is representing the bathymetry. Following the comment on the results and discussion section, all marine, perpendicular CRP profiles will be placed on a single figure, while the two land ERT profiles are put on a separate figure. «

Another important improvement would be to differentiate the water column over the aquifer. As this is a research about groundwater, the properties of the sea are not that needed in the figures and would help in the graphical interpretation of the results.

» It is important to understand that the water column is part of the inverse problem for CRP, in the sense that its resistivity is needed to solve it and has an influence on the obtained results in the aquifer. The larger the water column, the lower the sensitivity

in the aquifer. The bathymetry is indicated by the dotted white line. The water column conveys essential information, but we understand that removing the seawater makes the interpretation easier. We propose to do the following: the colour of the seawater layer will be uniform (green/blue) in figure 4, 5, and 6 and the resistivity of the layer is kept in figure 8. In line with the reorganization of the results section, the land ERT profiles will be placed on a separate figure, while the perpendicular CRP profiles (both low and high tide) are put on one plate. «

If there are specific areas of the cross-sections the authors want to refer to during the explanations, they can mark them in the figures to facilitate the link between text and figures.

» The tidal inlet, groundwater outflow, the Kortrijk Formation and other relevant elements will be indicated on the inversion profiles. «

I understand that the location of the cross-section in the figures corresponds with the spatial location, even if this is not clearly stated. The authors can add a reference point so the reader can know always how far is from the shoreline (i.e, distance to the high/low tide mark/dunes/).

» The low water line will be added as a reference point (0 m) on the figures with the land and marine ERT cross-sections. «

Also, all the profiles must have the same markers, for example, the dash lines in figure 4 are only in two of the cross-sections. In general, any improvement to give a better overview of the location of the cross-section would benefit the figures.

» We will make the necessary modifications. Also, the addition of a legend will partly resolve this confusion. «

It is not clear reading the figure caption of figure 7 what are the differences between A, B, C and D.

» These are horizontal slices of the subsurface with the electrical conductivity. The ap-

proximate depth of investigation given by the coil configuration of A, B, C, and D would be 0.9-1.8, 1.5, 2.1, and 6 m respectively. However, the depth at which the signal sensitivity is highest is somewhat ambiguous, since it also depends on the distribution of conductivity. The approximate depth values are not good indications in highly conductive conditions. Therefore, we cannot use them to refer to specific depth. Although for some people it would feel natural to use these values, they have no physical meaning here, and it is better to just provide the coil configurations. But to make the comparison in figure 7 easier, we will also add the approximate depths of investigation as pseudodepths. «

Figure 8 is difficult to follow even after reading the text (maybe because it is quite brief). A presentation of the results adapted to the purpose might be considered instead of presenting the data in the same format as in the analysis of the results (for example showing the difference in resistivity between inversion models).

» The fact that the three inversions look very similar is actually what we want to show. It gives us confidence about the inversion results. This remark is why we additionally show the depth of investigation (DOI). The zones with a higher DOI are the areas where there is a larger difference in resistivity between the inversions that use 0.1 and 10 times the reference resistivity, meaning that the confidence in the results is somewhat lower. We will try to better guide the reader through this figure. «

---

## Author Comment (AC2) · 30 Mar 2020

We thank the reviewer for his/her constructive comments. We understand that most of them are related to the interpretation of geophysical data for non-expert. We will try to clarify those as much as possible, but we cannot provide a comprehensive overview of geophysical method within a single paper of limited length. Below, our responses are indicated with "» «".

Abstract

Page 1 lines 18-24: This may be reduced/deleted as it is not really necessary and

lengthens the abstract unnecessarily.

» The fact that we use the robust apparent conductivity obtained by FDEM is a novelty in the research of coastal hydrogeology, it is therefore highlighted it in the abstract. But we will reduce this section. «

Introduction

Page 2 lines 17-34: As for non-geophysicists, it is extremely difficult to comprehend the suitability of each method for SGD detection as neither the method itself nor its potential/limitations are known. Listing them certainly shows that the authors have done a marvellous job in literature research, but it may be more advisable, for the sake of clarity, to spend some words on the methods in a general fashion (vertical resolution, drawbacks, uncertainties in the context of SGD) allowing the reader to better understand.

» Thank you for giving us insight from the perspective of a non-geophysicist. We try to give an overview of the previous uses of geophysics in this context, in order to show how our new methodology fills the gaps. We will try to summarize those studies to better underline their potential and limitations (e.g. resolution). It is however not our intention to provide basic details on the uses of these methods in the introduction as this strongly depends on the design of the system, we let this description to the methodology section. «

Methodology

3.1.3 Processing (electrical resistivity): If I understand correct, the authors did not apply any ground truth data (neither bathymetry, nor any drillings) to invert the raw ERT signals. Instead the three inversions are based on subjective thresholds?

» Bathymetry and seawater conductivity were acquired simultaneously to CRP (page 6 lines 2-3), these are important in the inversion of the marine ERT data because they strongly influence current flow paths in the subsurface. The measured bathymetry

is fixed during the inversion while the conductivity of the water layer is introduced as a reference value. The inversion can thus slightly deviate from this reference. It is not advised to fix it, because it can create artifacts in the inverted model. Due to logistical limitation no drillings are available on the beach and at sea (only in the dune). Those are not used for inversion but just for validation. The inversion stops based on convergence criteria typical for ERT applications. «

If so, and all results are based on relative terms, I would urge the authors (I raised a similar request during the results and discussion session), to clearly state (possibly in a separate subsection), which of the results can be stated with certainty given all uncertainties and relative terms/subjectivity.

» The assessment of the quality of a geophysical image is a very important research topic. It is important to understand that the inversion acts as a filter which tends to blur the image, so that the inverted resistivity is only an estimation of the true resistivity. For this reason, it is impossible to link an inverted resistivity value to a specific value of the salinity. However, in this case, we have two well identified extremes (freshwater in the dunes and seawater), together with a petrophysical model (Lebbe, 1978, 1999), so that we can interpret semi-quantitatively the variations of resistivity in terms of salinity classes. To validate the reliability of the inversion models, we show both the DOI and different inversions with/without reference models (figure 8). The DOI is an image appraisal tool used to indicate which portions of the image can be interpreted. If we impose different reference models during inversion, the observed patterns and resistivities in most zones (figure 8, e.g. A1 vs. A2 and A2 vs. A3) remain similar. This indicates that the resistivity structure are driven by the data and yield to low DOI value. Our DOI images show relatively low values everywhere, indicating we can trust the relative resistivity variations which occur in the subsurface. But we are careful in interpreting zones with a higher DOI (higher uncertainty). Both dedicated section of the manuscript will be modified to clarify those important points. «

Page 7 line 9: What does that mean? DOI <0.1 are more reliable than 0.2? Please

add this information and how the readers should interpret figures with DOI.

» The DOI is a widely used image appraisal tool in geophysics, this is why we did not add the corresponding equation. It is calculated as DOI=|log(rho(inv,1))-log(rho(inv,2))| / |log(rho(ref,1))-log(rho(ref,2))|. Where rho(inv) and rho(ref) are the inverted and reference resistivity. The logarithm of resistivity is used because it is the parameter inverted for. The higher the DOI, the more different are the results of two inversions and the less robust the ERT inversion results are. A threshold of 0.1 or 0.2 is commonly used by other authors, but this is subjective. Which is why we are showing both inversion results. More information will be provided on how to calculate the DOI and interpret figure 8. For additional information on the DOI, we also refer to Oldenburg, D.W., and Li, Y. 1999 [Estimating depth of investigation in dc resistivity and IP surveys, Geophysics, 64(2), 403-416, https://doi.org/10.1190/1.1444545] and others.

Page 7 line 25: Later on (pg10 L32) it is mounted on a sled. Please be consistent.

» It is actually mounted on a sled which is towed by a quad, this will be modified. «

Results and discussion

Page 8 lines 10-11: Why is TDS a topic here? It was never before mentioned. I suggest to remove it and to introduce section 4 differently.

» We agree with the reviewer and the result section should start with another sentence. The TDS is relevant because resistivity is only a proxy for the salinity, this section will therefore be transferred to the methodology. In this area, we have a good estimation of the formation factor of Archie's law (Lebbe, 1978), so resistivity can be interpreted as water salinity at the condition that the inversion models are sufficiently accurate. But the influence of inversion does not allow for quantitative interpretation in every zone. We know that some zones of the inverted image are less well resolved (they experience more smoothing, see also our answer below), meaning that some zones in the models indicated as brackish might be more salty or fresher. We will better clarify

that the water quality classes deduced from the models only give an estimation of the pore water quality. We will also specify the petrophysical relationship used to derive the classes (Archie's law for our study site). «

Page 8 lines 26-27: The entire text is hard to read and understand others than geophysicists as it is always difficult to interpret resistivity/ conductivity in light of complex and dynamic environments. Nevertheless, I am convinced the geophysics can help shedding light on SGD occurrence on a larger scale and even in the vertical dimension, which other methods cannot. To this end, it would be beneficial, if the authors can add more information to the figures (k1, where is pot. SGD, add a second color bar with likelihood of SGD instead of only resistivity/conductivity) to increase the instantaneous understanding.

» Thanks for this suggestion. Locations of (potential) fresh SGD will be highlighted on the figures. «

Page 9 lines 2-7: I encourage the authors to add all important terms (clay lens, saltwater lens, fresher water, freshwater) into to figure (Fig. 4) to clarify what they are referring to. Without knowing which values belong to which water component it is difficult to understand.

» Thanks for this suggestion. These terms will be added to the ERT and CRP profiles. «

Page 9 lines 20: Again, provided figures do not allow to follow the argumentation. Where is the low water line in Fig. 4 Please consider extensive improvements concerning the naming of terms in the figures to facilitate reading/understanding.

» Thanks for this suggestion. The low water line will be added as a reference point (0m) on the figures with the land and marine cross-sections. «

Page 9 lines 21-22: Why is that? What prove do the authors have to state such a process-based argument from mono-temporal recordings?

» We agree with the reviewer that this statement cannot be made from geophysical data only. We are biased by our overall knowledge of the process occurring here. We will rephrase "but the brackish groundwater is clearly pushed upwards" to "but the brackish groundwater is found relatively close to the seabed". «

Page 9 lines 31-32: This sentence perfectly proves what I stated before. Some of the statements given by the authors are pure assumptions when uncertainties are considered such as the different vertical resolutions and heterogeneities that influence that resolution. I would urge the authors to critically review the results in light of SGD detection. The main questions must be, what are the findings related to SGD that have a definitive certainty? (considering e.g. the multiplicity of methods with "same" result)

» We do not completely agree. These are not pure assumptions but interpretation accounting for the known limitations of inversion. This concept can be illustrated with a simple theoretical case. We start with a simple subsurface model which has four different resistivities, 0.5 (salt water), 5 (clay layer), 50 (fresh groundwater), and 500 $\Omega$m (unsaturated zone in the dunes) mimicking the situation encountered on our site (Figure 1_this_document, bottom). Forward modelling is applied by using RES2DMOD, simulating field data acquisition for this model. We obtain the following apparent resistivity pseudo-section (= data set) (Figure 1_this_document, top). Next, we process and invert the forward model exactly as we do our field data. The following inversion model is obtained (Figure 2_this_document, bottom). Here, you can clearly see that the zone in which there should be freshwater (see theoretical model) is not completely uniform because of smoothing. Since the resistivity is lowest where the saltwater lens is thickest. And this is exactly what was observed on profiles K1 and K0 (Figures 3 and 4_this_document). We will refer to Hermans & Paepen. "Combined inversion of land and marine electrical resistivity tomography for submarine groundwater discharge and saltwater intrusion characterization." Geophysical Research Letters (2020): e2019GL085877 where such a synthetic case is presented for more information. «

Page 10 lines 4-5: This line reads somewhat strange. Please consider a rewording/restructuring of the sentence.

». We'll rephrase it to: "since the groundwater system is least affected by anthropogenic effects in this part of the study area". «

Page 10 line 6: I assume the authors means salty, do they?

» Indeed, it is salty pore water. «

Page 10 line 10: I am afraid, it is not. Above the authors mentioned that all values >20 Om indicate fresh water. The maximum values in Fig 2 are <3. So, maybe I missed something, but it is not clear to me and I cannot follow the line of thought. Please, I strongly encourage the authors to add an axis/scale/bar/arrow to indicate in which part of Fig. 2 the authors see freshwater.

» We understand the confusion of the reviewer. The raw data are presented as apparent resistivity. The apparent resistivity is the value a homogeneous Earth should have to generate the exact same reading. It means that they are not only influenced by the aquifer resistivity but also (and mainly) by the low seawater resistivity. Only after inversion, which accounts for the effect of the low resistive seawater layer (by means of the bathymetry and seawater conductivity), we get an estimation of the true resistivity, you will see that the aquifer is brackish or fresh (Figure 5_this_document, is figure 2, left vs. figure 4). However, from the raw data, we can see an increase of this apparent resistivity. Assuming a constant influence of the seawater (what is true as long as its thickness remains similar), this must correspond to an increase in the aquifer resistivity, and thus a decrease in its salinity. This concept is well-known from geophysicists, allowing to draw qualitative conclusion from the raw data. We will clarify this in the manuscript. And the potential zone of discharge will be highlighted in figure 2. «

Page 10 lines 10-11: Is this so? Please keep in mind that I am a non-geophysicist, but primarily I would assume signals to be constant as long as nothing changes rapidly. The increase the authors show in Fig. 2 are rather smooth, so there is no rapid change.

SGD, in my eyes, would be rather rapid as it occurs at a certain e.g. sediment horizon. As soon as the device passes the horizon, the signal should show a rather rapid change, which I cannnot see in Fig. 2. So how can CRP be used as a rapid exploration technique?

» Similar to the previous question, you have to keep in mind that the raw data is highly influenced by the seawater layer. Its depth slowly decreases to the right of the graph (figure 2, left), leading to a slow increase of apparent resistivity. The increase is enhanced between 150 and 200 m, which is exactly where you find more brackish pore water in the aquifer (figure 4, K0HT). Additionally, you have to keep in mind that an apparent resistivity corresponds to a combination of four electrodes. The distance between those electrodes are 15 m, so that the minimum volume investigated by CRP is at least 60 m long (and gets longer when larger electrode spacing are used, what is necessary to investigate the aquifer). Therefore, the transition in the raw data will always be smoothed as this investigated volume acts as a filter. This is another reason why inversion is needed. This rapid change, as you expect in the signal, is attenuated on the raw data by the influence of the seawater and investigated volume. This will be explained in the manuscript. «

Page 10 lines 12-13: If the authors would scan a line parallel to the beach, but the "leakage" or discharge zone is more offshore, the authors would not see the discharge itself but just the freshwater. So yes, this could be used to see how far freshwater may extend offshore but, a) the scanline should then be very close to the shoreline to not miss SGD close to the shore (which by the way may be difficult due to the a certain depth either the boat or the device may need) and b) to really "see" groundwater discharging, the scanline should be directly above it. If I am mistaken, please correct me, otherwise I suggest to alleviate the very strong statement.

» The reviewer is right about parallel profiles. Perpendicular profiles do not suffer from this limitation. However, we do not claim that CRP can be used to detect fresh SGD everywhere, but it can be used as a quick exploration technique (when you use the

raw data) to find zones worth investigating in terms of nutrient/contaminant leakage by SGD. Fresh groundwater discharge does not have to be a discrete outflow, it is often quite diffuse. Making the actual detection of freshwater discharge at the seabed is quite difficult with resistivity measurements. The presence of fresh pore water in the marine aquifer is already proof of fresh groundwater discharging, since it migrates through the sediments towards the seawater due to the buoyancy effect. Also, the upper layer of the marine aquifer is sometimes filled with brackish pore water, making it problematic to see fresh discharge at the seabed surface. «

Page 10 line 16: Please try to avoid any terms that tangent a temporal dimension unless you can prove it. I mentioned it before, the authors record mono-temporal signals which do not allow any statements on temporal dimensions.

» This difference has nothing to do with temporal variation, but with the spatial variation (compared to the groundwater extraction site) and the effect of the saltwater lens thickness on the inversion smoothing (see answer to: Page 9 lines 31-32). We know from modelling studies that in this context, a higher groundwater flux will lead to a thinner saltwater lens, Vandenbohede and Lebbe, 2006a, this will be explained in the manuscript. «

Page 11 line 7: This is likely, but, if I am not mistaken, it can also be due to different methods, a change in heterogeneity due to wave actions/storms etc. Please relativize the authors statement if I am correct.

» Both methods where the same, FDEM. We only used two different devices, with different depths of investigation. Storm and wave action might indeed have an effect on the heterogeneity on the lower beach, but both zones of higher resistivity are related to fresh discharge. This is based on comparison with the (marine) ERT profiles, the zone does not move, only the intensity of the discharge changes. «

Page 11 lines 8-9: This does not support the hypothesis of less discharge (if the signal is really discharge based). The main questions in this context would be, what are the

travel times? What are the groundwater ages? Both would support the statement given by authors but without, it is another assumption and no reasoning.

» We partly disagree with the reviewer. It is true that our interpretation is partly biased by our knowledge of the field. Modelling results in the area show that a decrease in the flux will induce an increase in the salinity of the discharge water, what is confirmed by our geophysical data. We will rephrase the sentence. This statement will be confirmed by planned flux monitoring. «

Conclusions

Page 12 lines 15-16: From my perspective, this statement should be deleted.

» This sentence will be removed. «

Figures

Figure 3: Why does the color bar include values <100 when apparently neither the left nor the right shows these values? As a consequence the reader may see smaller signal variations.

» We choose to use the same colour scale as the resistivity profiles, to help in their correlation. The right image (figure 3) is actually the same as figure 7A, which allows you to see a more detailed conductivity variation. Figures 3 and 7 will be placed in sequence, due to the reorganization of the results section, making this more clear. «

Figure 4: I would suggest to add to features: 1. A scale showing the likeliness of SGD 2. A meter scale beneath all profiles that facilitates the understanding of what the authors talk about (e.g. pg9 L18 // pg10 L7) that includes information on low water line. Besides, are the abs. error statements in the captions needed?

» Zones with (potential) fresh SGD will be highlighted in the profiles and a meter scale will be provided, with the low water line as a reference (0 m). The absolute errors will be provided in the supplements, since it is good practice in geophysics to communicate

the final inversion error. «

Figure 8: This figure is overloaded and almost impossible to understand for non-geophysisists. Please rework or/and consider moving it to the auxillary data.

» We will try to better guide the reader through this figure, using a dedicated section in both the methodology and results section. As explained above, this figure is essential to assess the reliability of the inversion and should thus be kept within the main text. «
* * *
Ps.Z

Blocks (Wenner Alpha array)

Apparent Resistivity Pseudosection

Resistivity in ohm.m

Resistivity model

**Fig. 1.**

**Fig. 2.**

grad8tri_1

Measured Apparent Resistivity Pseudosection

Calculated Apparent Resistivity Pseudosection

Depth   Iteration 7 Abs. error = 1.73 %

Inverse Model Resistivity Section

Resistivity in ohm.m

0.312  0.883  2.50  7.07  20.0  56.6  160  453

Unit electrode spacing 5.00 m.

**Fig. 3.**

grad8tri_1

Measured Apparent Resistivity Pseudosection

Calculated Apparent Resistivity Pseudosection

Depth   Iteration 7 Abs. error = 2.8 %

Inverse Model Resistivity Section

Resistivity in ohm.m

Unit electrode spacing 5.00 m.

**Fig. 4.**
Apparent resistivity (Ωm)

Potential zone of discharge

Fig. 5.

---

## Editor Comment (EC1) · Gerrit H. de Rooij (Editor) · 1 Apr 2020

The reviewers agree that the paper presents material worth publishing but have concerns about the presentation. The reviews offer many suggestions to improve the flow of the paper, and referee 1 offers numerous additional references that may help positioning the paper in the existing literature.

Referee 2 added a supplement that raises valid questions about the interpretation if the data by the authors. One of the issues raised by the referee is the fundamental problem that arises when data collected in a very narrow time window are used to draw conclusions about time-variant processes. I can imagine that this is a problem

that occurs frequently in geophysics, because data are frequently collected in short, intensive campaigns. I would welcome it if the authors would discuss this issue in the text and clarify which conclusions they can confidently draw from the data, and which inferences are more speculative (should they choose to keep them in).

I agree with the reviewers that the structure and organisation of the paper needs to be improved. The responses of the authors indicate that the points the reviewers raised are taken and they know how to remedy them.

Overall, I believe the reviews are such that a revision of the paper is warranted. No new data are necessary, but elements of the data analysis, many details in the text and the figures, and the overall organisation of the paper require substantial improvement to make the paper more accessible.

I therefore recommend a major revision.

Sincerely,

Gerrit de Rooij

Editor

---

## Author Response (AR1)

[revised manuscript text omitted]

heeft verwijderd: 5

heeft verwijderd: it

heeft verwijderd: (Figure 5)

heeft verwijderd: Note that t

heeft verwijderd: at high tide is

heeft verwijderd: RSGD

heeft verwijderd: in the discharge

heeft verwijderd: and

heeft verwijderd: s

heeft verwijderd: e.g. K1 (Figure 8, A),

heeft verwijderd: s

heeft verwijderd: , e.g. K1$_{HT}$

heeft verwijderd: C1

heeft verwijderd: high

heeft verwijderd: the discharge zone

heeft verwijderd: s, e.g. K1$_{LT}$

heeft verwijderd: B1

heeft verwijderd: discharge

heeft verwijderd: 7

heeft verwijderd: with CMD3

heeft verwijderd: 7

heeft verwijderd: discharge

heeft verwijderd: using HCP4

heeft verwijderd: 7

heeft verwijderd: with HCP1 and PRP4

heeft verwijderd: 7

heeft verwijderd: a

[revised manuscript text omitted]

---

## Author Response (AR2)

[revised manuscript text omitted]

**Met opmerkingen [MP1]:** Surfer automatically gives weightings to the points, the default settings for natural neighbour interpolation were used.

**3.3 Interpretation**

The inversion of resistivity data acts as a filter which tends to blur the obtained image: the inverted resistivity is only an estimation of the true resistivity. For this reason, it is difficult to directly relate an inverted resistivity value to a specific value of the salinity or total dissolved solid content (TDS). However, in this case, we have two well-identified extremes (freshwater in the dunes and seawater), together with a petrophysical model developed by Lebbe (1978, 1981, 1999) for the Western Belgian coastal plain, allowing a semi-quantitative interpretation of resistivity. Archie's law (Archie, 1942) is used to estimate the pore water resistivity ($\rho_w$):

$$\rho_w = \frac{\rho_b}{F} \quad (2)$$

For sandy sediments at 'De Westhoek', Lebbe (1981) estimated the formation factor ($F$) to be 3.2. The bulk resistivity ($\rho_b$), is deduced from ERT/CRP profiling and FDEM mapping (given the scale of measurement, we assume that the rECa can approximate $\rho_b$). Here, we propose a semi-quantitative interpretation in salinity classes based on relative variations in resistivity (and conductivity). Due to measurement errors, resolution, and inversion constraints, deducing the effective TDS would only be possible with a specific calibration of geophysical measurements based on ground truth data. We distinguish three main water quality classes (salt, brackish, and fresh) to interpret the geophysical data in the specific study area (De Moor and De Breuck, 1969).

**heeft verplaatst (invoeging) [1]**

**heeft verwijderd:** $\rho_w$ is

**heeft verwijderd:** where t

**heeft verwijderd:** and

**heeft omhoog verplaatst [1]:** $\rho_w$ is the pore water resistivity

**4 Results and discussion**

In the following paragraphs, we present the data and results for each of the used methods, starting with FDEM, then land and marine resistivity measurements. In the east of the study area, FDEM data were acquired during two different seasons, allowing to investigate seasonal variability of the fresh groundwater discharge. By comparing our results to previous observations, we can see how the fresh-/saltwater distribution has changed since the 1980s, due to a decrease of pumping in the nearby groundwater extraction facility. In the following, we will use a unique colour scale to describe the results in all figures (except in Figure 4): freshwater occurs in zones with a resistivity higher than approximately 20 Ωm (blue), salt water has a resistivity lower than roughly 2.5 Ωm (red), and brackish water leads to intermediate resistivities (light brown to light blue). In the case of the Belgian coast, the strong tides play a role allowing salt water to penetrate in the shallow sediment, and making the detection of FSGD - based on resistivity/conductivity - more difficult. The discharge zone will thus not be necessarily characterized by freshwater but rather by brackish water.

**heeft verwijderd:** East

**4.1 FDEM**

We started our investigation with the CMD-MiniExplorer mapping at the location of one of the profiles of Lebbe (1981), K1, located at 1 km east from the French-Belgian border. The CMD-MiniExplorer mapping (Figure 3, right, and Figure 4, A.) identifies the presence of FSGD close to the low water line, at K1, indicated by a decrease in the electrical conductivity. The

5  FDEM data clearly demonstrates a zone over 100 m wide. The outflowing water is very brackish to moderately salt, the conductivity is roughly between 350 and 650 mS m$^{-1}$, due to a mixture of discharging fresh groundwater and seawater that infiltrated on the beach during high tide.

To investigate lateral variation between the western and eastern part of the study area, the intertidal zone was also mapped with the DUALEM at the vicinity of the low water line. This map clearly demonstrates a northward shift of the zone of

10  discharge from east towards west (Figure 4, B.-D.). From approximately 300 m east of the border, the discharge zone is no longer visible from the EM data since the discharge is located below the low water line. The DUALEM was dragged on the beach in a sled, making the quality of the data higher compared to the MiniExplorer, which was carried by hand, making it difficult to maintain a constant height above the surface during mapping. The water conductivity, obtained with the DUALEM-421S, in the discharge zone corresponds to a brackish water type, confirming the mixing of freshwater with salty water.

15  ### 4.2 Land ERT

Interpreted on its own, the conductivity variations observed with FDEM could be related to lithological heterogeneities or the presence of near-surface features. Therefore, long land ERT profiles (Figure 5) covering the entire zone between the low water line and the dune area were collected. In the east (Figure 5, K1), we identify the freshwater aquifer located in the dune area. The brackish water observed at -10 mTAW corresponds to remnants of seawater infiltration during the flooding of an artificial

20  tidal inlet, which started in 2004 and ended a few years ago. The downward movement of the denser seawater is hindered by the presence of a local clay lens, a process that was closely monitored in previous studies (Vandenbohede et al., 2008b; Hermans et al., 2012). Along this profile, there is a large saltwater lens on the beach. Underneath, freshwater is flowing from the dunes towards the North Sea. The freshwater mixes with the salt water, leading to the discharge of salt to brackish water during low tide on the lower beach. This is clearly visible on the land profile, close to the low water line, where fresher water

25  is seen near the surface, which corresponds to the zone identified by FDEM measurements.

In the centre of profile K1 (Figure 5), the groundwater appears to be brackish. Here, the thickness of the salt water lens is maximum (15 m), while the bottom boundary of the aquifer (thick clay layer) is located between -25 and -30 mTAW. The lower resistivity is likely a result from the smoothness constraint inversion combined with the lower resolution at depth as the more resistive freshwater is bounded by two conductive layers, leading to a lower inverted resistivity (Hermans & Paepen,

30  2020).

There is a similar situation 1 km to the west, at the Belgian-French border (Figure 5, K0), where freshwater moves underneath a saltwater lens from the dunes towards the sea. Nevertheless, a striking difference on this profile is the absence of freshwater

heeft verwijderd: East

heeft verwijderd: which is

heeft verwijderd: East

heeft verwijderd: West

heeft verwijderd: East

heeft verwijderd: East

heeft verwijderd: West

[revised manuscript text omitted]